# Single-molecule digital sizing of proteins in solution

Georg Krainer [1,2,11] ✉, Raphael P. B. Jacquat [2,11], Matthias M. Schneider [2,3,11], Timothy J. Welsh[2,11], Jieyuan Fan[2,4,11], Quentin A. E. Peter[2], Ewa A. Andrzejewska [2], Greta Šneiderienė [2], Magdalena A. Czekalska [2], Hannes Ausserwoeger [2], Lin Chai[2], William E. Arter[2], Kadi L. Saar [2], Therese W. Herling [2], Titus M. Franzmann [5], Vasilis Kosmoliaptsis [6,7,8], Simon Alberti [5], F. Ulrich Hartl [3,9], Steven F. Lee [4] & Tuomas P. J. Knowles [2,10] ✉

The physical characterization of proteins in terms of their sizes, interactions, and assembly states is key to understanding their biological function and dysfunction. However, this has remained a difficult task because proteins are often highly polydisperse and present as multicomponent mixtures. Here, we address this challenge by introducing single-molecule microfluidic diffusional sizing (smMDS). This approach measures the hydrodynamic radius of single proteins and protein assemblies in microchannels using single-molecule fluorescence detection. smMDS allows for ultrasensitive sizing of proteins down to femtomolar concentrations and enables affinity profiling of protein interactions at the single-molecule level. We show that smMDS is effective in resolving the assembly states of protein oligomers and in characterizing the size of protein species within complex mixtures, including fibrillar protein aggregates and nanoscale condensate clusters. Overall, smMDS is a highly sensitive method for the analysis of proteins in solution, with wide-ranging applications in drug discovery, diagnostics, and nanobiotechnology.

Proteins form the molecular machinery of life and accomplish their biological function by interacting with other proteins as well as by assembling them into biomolecular complexes and higher-order structures[1]. The characterization of proteins is thus a key objective in many areas of biological and biomedical sciences, both for understanding the normal functional behavior of proteins and for elucidating aberrant processes and interactions that can lead to dysfunction and disease[2–5]. Proteins also serve as important therapeutic targets in

[1]Institute of Molecular Biosciences (IMB), University of Graz, Humboldtstraße 50, 8010 Graz, Austria. [2]Centre for Misfolding Diseases, Yusuf Hamied Department of Chemistry, University of Cambridge, Lensfield Road, Cambridge CB2 1EW, UK. [3]Department of Cellular Biochemistry, Max-Planck Institute of Biochemistry, Am Klopferspitz 18, 82152 Martinsried, Germany. [4]Yusuf Hamied Department of Chemistry, University of Cambridge, Lensfield Road, Cambridge CB2 1EW, UK. [5]Center for Molecular and Cellular Bioengineering, Biotechnology Center, Technische Universität Dresden, Tatzberg 47/49, 01307 Dresden, Germany. [6]Department of Surgery, University of Cambridge, Addenbrooke's Hospital, Hills Road, Cambridge CB2 0QQ, UK. [7]NIHR Blood and Transplant Research Unit in Organ Donation and Transplantation, University of Cambridge, Hills Road, Cambridge CB2 0QQ, UK. [8]NIHR Cambridge Biomedical Research Centre, University of Cambridge, Hills Road, Cambridge CB2 0QQ, UK. [9]Munich Cluster for Systems Neurology (SyNergy), Feodor-Lynen-Str. 17, 81377 Munich, Germany. [10]Cavendish Laboratory, Department of Physics, University of Cambridge, JJ Thomson Ave, Cambridge CB3 0HE, UK. [11]These authors contributed equally: Georg Krainer, Raphael P. B. Jacquat, Matthias M. Schneider, Timothy J. Welsh, Jieyuan Fan. ✉e-mail: georg.krainer@uni-graz.at; tpjk2@cam.ac.uk

drug discovery and clinical diagnostics, and are utilized as nanoscale building blocks in bionanotechnological applications[6–8]. A rigorous analysis of proteins and protein assemblies is therefore essential for advances in therapeutics development and the discovery of biomaterials[9–11]. In particular features such as the molecular size of proteins and protein complexes, the strength of their interactions in terms of pairwise dissociation constants, and their assembly and oligomerization states are key parameters providing a heightened understanding of the biological function, malfunction, and design of proteins[12–14]. While such biophysical characterization of proteins has become routine, there still remain major hurdles that have not been addressed and which pose significant challenges for currently available biophysical approaches.

One particular challenge in the characterization of proteins lies in their inherent compositional complexity and heterogeneity. Protein systems often exist as mixtures of heterogeneous components, and exhibit polydispersity in terms of size and abundance. However, most classical biophysical approaches perform best when applied to pure homogeneous samples, and methods developed to enable the profiling of heterogeneous mixtures, such as gel filtration, electrophoresis, mass spectrometry, and surface plasmon resonance, pose significant problems[15–18]. These approaches are generally reliant on separation media, immobilization, or transferring the molecules from the solution phase into the gas phase, all of which may modify the distribution of sizes and thus make it challenging to relate the functional state of a protein to that probed under native solution conditions. While methods capable of studying molecules in solution exist, such as analytical ultracentrifugation, isothermal titration calorimetry, and static and dynamic light scattering, they generally consume large amounts of protein and often require concentrations exceeding the biologically relevant range[19–21]. Many proteins are, however, present at low concentrations in biological media or samples are often available only in limited amounts. Therefore, ultrasensitive approaches that can size proteins and resolve heterogeneous mixtures of proteins at very low concentrations directly in solution are much sought after.

Single-molecule detection methods have emerged as powerful tools in addressing these challenges. They provide rich insights into subpopulations and compositional complexities and facilitate resolving heterogeneous protein systems at ultra-low concentrations. Techniques such as single-molecule fluorescence spectroscopy have advanced protein and protein complex sizing through correlation[22,23] and brightness analysis[24,25] of fluorescence signals or by counting fluorophore photobleaching steps both in vitro on purified protein samples and in live cells[26,27]. Despite their widespread use, these methods encounter challenges like the need for calibration to determine absolute sizes, difficulties in quantifying larger assemblies due to photophysical constraints, and complications linked to overparameterization in the fitting of higher-order assemblies[25,28,29]. Additionally, the advent of various single-molecule super-resolution microscopy techniques has enabled nanoscale visualization of proteins, uncovering ultrastructural details of protein assemblies and interactions both in vitro and in live cells, and offering quantitative insights into protein number, size, distribution, and spatial organization[30,31]. However, these techniques often require complex sample preparation and labeling steps and involve sophisticated image reconstruction and data analysis procedures that can introduce artifacts and biases. More recently, methods like mass photometry[32], or also known as interferometric scattering microscopy[33–35], have gained prominence. These methods enable label-free detection of proteins and protein assemblies and the quantification of dissociation constants in protein complexes[36–38]. While beneficial in many contexts, mass photometry can face challenges with specificity in complex samples due to the absence of labels and is more suited to characterizing larger proteins and assemblies. Consequently, there is an ongoing need to develop methodologies that can mitigate some of these challenges.

In this work, we present a single-molecule microfluidic approach, termed single-molecule microfluidic diffusional sizing (smMDS), for the characterization of the sizes, interactions, and assembly states of proteins in solution. smMDS utilizes single-molecule fluorescence detection to measure the molecular diffusivity of individual proteins and their assemblies within microchannels. By incorporating a confocal fluorescence readout functionality into a microfluidic platform based on the principles of microfluidic diffusional sizing (MDS), smMDS records diffusion profiles through digital counting of individual molecules. This approach enables calibration-free and absolute measurement of protein hydrodynamic radii at the single-molecule level, even in complex, multicomponent mixtures. In the following, we introduce the smMDS platform and elaborate its working principle and experimental implementation, highlighting its capability for sizing single proteins and complexes with sensitivities down to femtomolar levels. We show that smMDS proves effective in quantifying protein interaction affinities at the single-molecule level and resolving diverse assembly states of protein oligomers. Furthermore, we apply smMDS in characterizing aggregate assemblies in multicomponent protein mixtures and sizing nanoscale clusters in protein condensate systems. We also apply smMDS to characterize complex aggregate assemblies in multicomponent protein mixtures and for the sizing of nanoscale clusters in protein condensate systems. Overall, we show that smMDS represents a versatile, in-solution approach for digitally characterizing protein sizes, interactions, and assembly states. We anticipate that smMDS will have widespread applicability in the biological and biomedical sciences for the discovery of biomolecular mechanisms underpinning protein function and malfunction. Furthermore, smMDS holds great potential for the analysis of protein interactions in drug discovery, clinical diagnostics and nanobiotechnological applications.

## Results
### Rationale for developing smMDS
Micron-scale measurements of molecular diffusivity have proven to be a versatile and sensitive approach for probing the molecular sizes of proteins, their interactions, and assembly states in solution[39–41]. In particular, microfluidic diffusional sizing (MDS) has become an attractive, quantitative method for characterization of proteins and protein complexes under native solution conditions[42–54]. MDS exploits the unique features of laminar flow in the microfluidic regime and measures the diffusive mass transport of molecules across co-flowing sample and buffer streams within a microchannel[41,42]. By monitoring the diffusive spreading of analyte molecules at different downstream channel positions and analyzing the recorded diffusion profiles with advection–diffusion models, the sizes of analyte molecules can be quantified in terms of their hydrodynamic radii $R_h$. Importantly, MDS offers the advantage of being calibration-free, as it directly retrieves $R_h$ from model fitting, thereby eliminating the need for external calibration standards. MDS can further probe the formation of biomolecular interactions and the assembly of analyte molecules into higher order structures by monitoring the increase in size associated with complex formation, and can retrieve binding affinities (i.e., dissociation constants $K_D$) through measurement of binding curves. However, despite the versatility of the approach, current implementations of MDS and other similar microfluidic methods are limited in their ability to resolve compositional heterogeneities. This is because they rely on ensemble readouts that average out the signal, making it difficult to determine the size distribution of different species, thus yielding only an average $R_h$ for mixtures of differently sized species[42,55,56]. This limits information content, particularly when studying heterogenous, multicomponent systems. Additionally, detection sensitivities for MDS and other similar sizing techniques are in the nanomolar to

micromolar range, which hampers ultrasensitive protein detection at concentrations in the often desirable pico- to femtomolar range[42,55,56].

A more sensitive detection method, like confocal fluorescence detection, could offer an effective solution to these challenges. This technique enables ultra-sensitive detection at the single-molecule level[57–59], allows for direct digital readouts by counting individual proteins and protein complexes in solution[60], and can be seamlessly integrated with microfluidic platforms[60–64], thus setting the stage for developing a single-molecule digital sizing method. We therefore reasoned that combining single-molecule fluorescence detection with diffusivity measurements in microchannels would create a robust platform for sizing proteins at the single-molecule level in a calibration-free manner. Such a platform would be ideally suited for characterizing heterogeneous and multicomponent protein systems directly in solution.

## Working principle and experimental implementation of smMDS

Figure 1a illustrates the working principle and experimental implementation of smMDS. smMDS measures the molecular diffusivity of analyte molecules within a microfluidic chip. It operates based on the principles of MDS[42] and probes molecular diffusivity by flow-focusing an analyte stream between two auxiliary buffer streams within a microfluidic chip and then observing the diffusive spreading of analyte molecules to either side of the microfluidic channel as they travel downstream (see Methods). Because different positions along the channel correspond to different diffusion times, the tracking of the diffusive broadening of species at different channel positions allows calibration-free quantification of the diffusion coefficient $D$ and, thus, extraction of the size of analyte molecules in terms of $R_H$[42]. Experimentally, smMDS measurements are conducted by introducing a sample containing fluorescent protein and buffer into the sizing chip and monitoring the micron-scale diffusive mass transport of molecules across the channel as they flow downstream the channel. Fluid flow in the channel is controlled by applying a negative pressure at the device outlet with a syringe pump (see Methods for details).

Detection in smMDS is achieved using a high-sensitivity laser confocal fluorescence microscope functionality incorporated into the microfluidic platform (see Methods). By scanning the confocal volume across the microfluidic chip at the mid-height of the channel perpendicular to the flow direction (Fig. 1a), fluorescence from passing analyte molecules is recorded. The scan trajectory is chosen such that various positions along the channel are probed, including positions that are close to the nozzle where the sample stream meets the co-flowing buffer medium, and others, further away, downstream of the channel. In our implementation, the four innermost channels of the device are scanned to obtain diffusion profiles. These selected

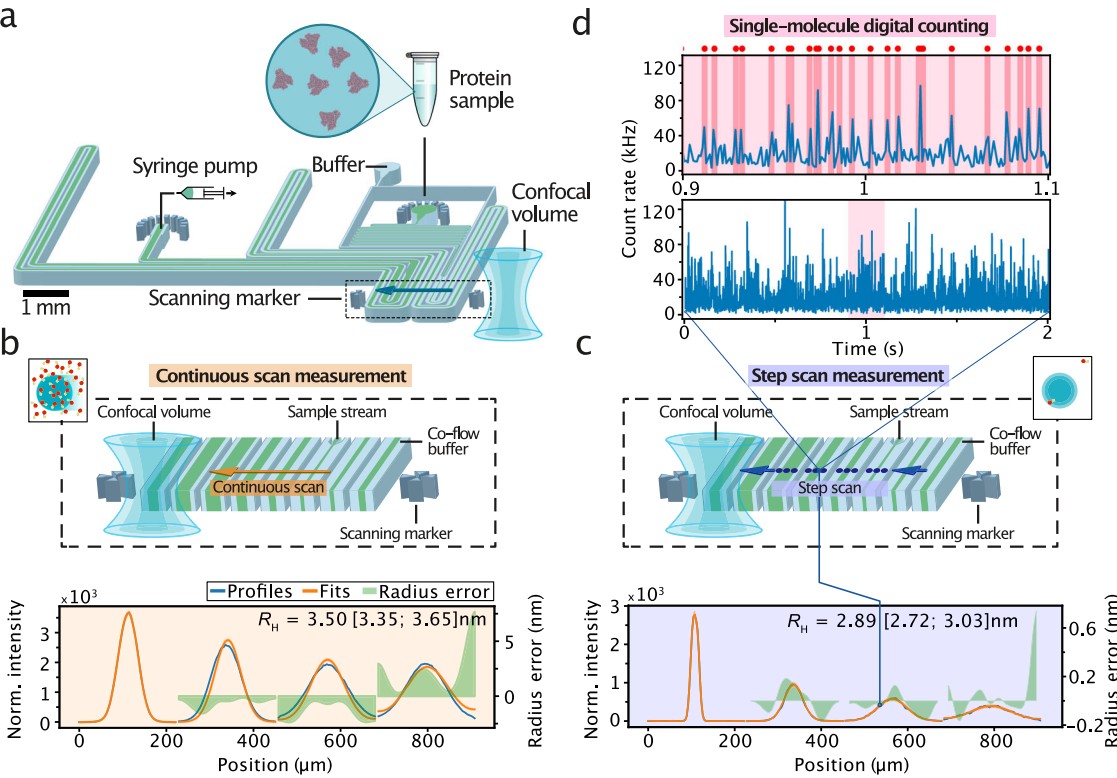

**Fig. 1 | Working principle and experimental implementation of smMDS.**
**a** Schematic of the microfluidic chip design and integrated confocal scanning optics. The most relevant components are depicted. The dashed box highlights the scan region. The arrow indicates the scan trajectory across the four innermost channels. **b** Principle of continuous scan measurements. The confocal detection volume is moved at a constant speed across the microfluidic device, enabling the recording of diffusion profiles from direct intensity readouts. This mode enables recording of diffusion profiles under ensemble conditions. An exemplary diffusion profile from a continuous scan measurement of human serum albumin (HSA) at 100 nM is shown. Diffusion profiles are shown as blue lines, experimental fits as orange lines, and the local radius errors as green bands. Extracted hydrodynamic radii $R_H$ [with errors] are given as an inset. The local radius error is calculated as the difference between the hydrodynamic radius derived from the global fit and that obtained from the best matching profile at that specific position. The error range for $R_H$ is derived from the global fit, determined through a Taylor expansion of the least-square fit and through error propagation (see Supplementary Note 1 for details). **c** Principle of step scan measurements. The confocal detection volume is moved in a stepwise manner across the device, collecting data at defined positions with each step for a certain period of time in the form of time traces (see panel d). This mode enables detection of individual molecules and the creation of diffusion profiles from single-molecule digital counting. An exemplary diffusion profile from a step scan measurement of α-synuclein at 10 pM is shown. **d** A single-molecule time trace (lower panel) as obtained from a step scan measurement is shown. The time trace in the upper panel is a zoom-in view of the red shaded area in the lower panel. Red dots and highlighting indicate bursts detected by the burst-search algorithm. The bin time is 1 ms in all traces.

channels cover a wide range of distances and time points along the channel's length. This enables the analysis of biomolecular analytes with $R_h$ ranging from less than 1 nm to greater than 100 nm, paralleling the established range in standard MDS experiments[42,47,65,66]. The scan trajectory of the confocal volume in x,y,z-direction is set through two scan markers integrated within the microfluidic chip adjacent to the channels.

Scanning is conducted in two modes, by either continuously moving the confocal volume through the chip (Fig. 1b), or by moving the observation volume along the same trajectory in a stepwise manner, collecting data at defined positions with each step (Fig. 1c). In continuous scan mode, diffusion profiles are rapidly acquired from direct intensity readouts. In this process, the confocal volume is moved through the device at a constant scan speed (tens of µm/s) and the fluorescence intensity from analyte sample flowing through the confocal volume is recorded. This allows swift recordings of diffusion profiles under ensemble conditions, that is, at concentrations where many molecules are present in the confocal volume (i.e., typically at concentrations greater than tens of pM). An exemplary diffusion profile obtained from a continuous scan experiment is shown in Fig. 1b. To extract $R_H$, the recorded diffusion profiles are analyzed using an advection–diffusion model (see Methods). This process involves fitting the experimental profiles to a set of simulated profiles generated through numerical simulations. Using a least-squares error algorithm, the best-matching simulated profile is identified via fitting to extract $D$ and retrieve $R_H$ using the Stokes-Einstein relationship (Fig. 1b).

In step scan mode, diffusion profiles are generated from time trace recordings along the scan trajectory. An exemplary diffusion profile obtained from a step scan experiment is shown in Fig. 1c. In this process, the confocal volume is parked at various positions along the trajectory, and fluorescence signals from analyte flowing through the confocal volume are recorded for a certain period of time. Typically, between 200 to 400 scan steps across the chip are performed from start to end position and 2- to 4-second-long fluorescence traces are recorded at each position. Importantly, due to the high sensitivity of confocal detection, measurements in step scan mode enable the detection of individual molecules and, thus, the creation of diffusion profiles from single molecule counting (Fig. 1d). Hereby, bursts of fluorescence corresponding to the passage of single molecules through the confocal volume are recorded at each channel position. To estimate the number of molecules at each scanned position, a burst-analysis algorithm is employed (see Methods). This algorithm uses a combined maximum interphoton time ($IPT_{max}$) and minimum total number of photons ($N_{min}$) threshold criterium to extract single-molecule events from the recorded time trace at each position (see Methods). This approach has been shown to enable effective discrimination between photons that originate from single fluorescent molecules and those that correspond to background, thus allowing individual molecules to be counted directly, that is, in a digital manner[60]. From the detected number of molecules at each position, diffusion profiles are then created by plotting the number of counted molecules as a function of chip position. Extraction of $R_H$ is done analogously as described above for continuous scan experiments by fitting the experimental diffusion profiles with our advection–diffusion analysis model. Figure 1c depicts fits and extracted $R_H$ values for the example data set.

## Sizing of proteins from bulk to single-molecule conditions by smMDS

In a first set of experiments, we sought to evaluate the sensitivity of smMDS and demonstrate its capability to determine protein size from bulk to single-molecule conditions. To this end, we labeled human serum albumin (HSA) with a fluorescent dye (Alexa 488) (see Methods and Supplementary Table 1) and performed concentration series measurements, both in continuous and step scanning mode, by

varying the HSA concentration. The recorded diffusion profiles of the series are shown in Fig. 2b, c.

In the range from 1 µM down to tens of pM of HSA (Fig. 2b), sufficient molecular flux of HSA protein molecules allowed for the recording of diffusion profiles from continuous scan experiments. The obtained profiles show the characteristic broadening due to diffusion of molecules along the channels. Narrow peaks are observed at channel positions close to the nozzle where the sample meets the carrier medium, and peaks broadened as we probed further downstream channel positions. Modeling of the diffusion profiles using our advection–diffusion analysis approach yielded excellent fits. Extracted $R_H$ values (Fig. 2d and Supplementary Table 2) were amongst all concentrations, within error, in excellent agreement with previously reported values for HSA[67–70], demonstrating the robustness and accuracy of the approach. Of note, in some cases, the diffusion profiles display a minor offset from the channel centers. We attribute this to potential small imperfections upstream in the flow path, which can slightly shift the peak positioning. However, the global fitting procedure we utilize for extracting diffusion constants is resilient towards anomalous shifts that occur within channels, ensuring that the overall analysis remains robust and accurate.

As we approached the single-molecule regime (i.e., at and below 20 pM) (Fig. 2c), we performed step scan measurements by moving the confocal volume in a stepwise manner across the channels. We observed bursts of fluorescence corresponding to the passage of single HSA molecules through the confocal volume (Fig. 2c, top panels). Using the burst-search algorithm (see Methods), we extracted single-molecule events from the recorded time traces at each position (Fig. 2c, top panels) and created diffusion profiles by plotting the number of counted molecules as a function of chip position (Fig. 2c). In this way, we obtained diffusion profiles from digital counting for HSA from 20 pM down to 100 fM. Remarkably, also in this regime, by applying our advection–diffusion analysis, we obtained excellent fits of the experimental data. We retrieved $R_H$ values that were in agreement with previously reported values for HSA[67–70] and within an error margin of 10% (Fig. 2d and Supplementary Table 2). Of note, HSA exists in an equilibrium of monomers and low-order oligomers, which are populated with decreasing relative abundance[32,71]; hence $R_H$ values reported are weighted averages reflecting this distribution of the monomeric and oligomeric forms (for a deconvolution analysis see Fig. 5). Overall, the results shown here demonstrate that smMDS provides accurate size information over a broad range of concentrations and enables ultrasensitive sizing of proteins even in the picomolar to femtomolar concentration regime. We also evaluated the influence of parameter selection in the single-molecule regime on the extracted sizes (Supplementary Fig. 1) and found that a wide range of burst selection parameters, that is, varying thresholds for $ITP_{max}$ and $N_{min}$ (see Methods), yielded expected size information, supporting the robustness of the approach.

To compare the sensitivity of the smMDS technique to conventional MDS measurements, we conducted experiments utilizing fluorescence widefield imaging (Supplementary Fig. 2). We performed concentration series measurements by varying the HSA concentration starting at 1 µM of labeled HSA and then gradually decreasing the protein concentration down to 100 and 50 nM. Image analysis yielded a clear profile only for the measurements at 1 µM and 100 nM protein concentration, and the expected size for HSA could only be recovered for the measurement at 1 µM protein. Notably, the measurement at 50 nM HSA yielded a featureless profile that could not be fitted and, hence, no $R_H$ could be determined. This shows that conventional MDS experiments are limited to concentrations in the tens of nanomolar range. In comparison, the high sensitivity and digital detection capabilities afforded by smMDS allows measuring the size of proteins down to femtomolar concentrations, thereby extending the sensitivity

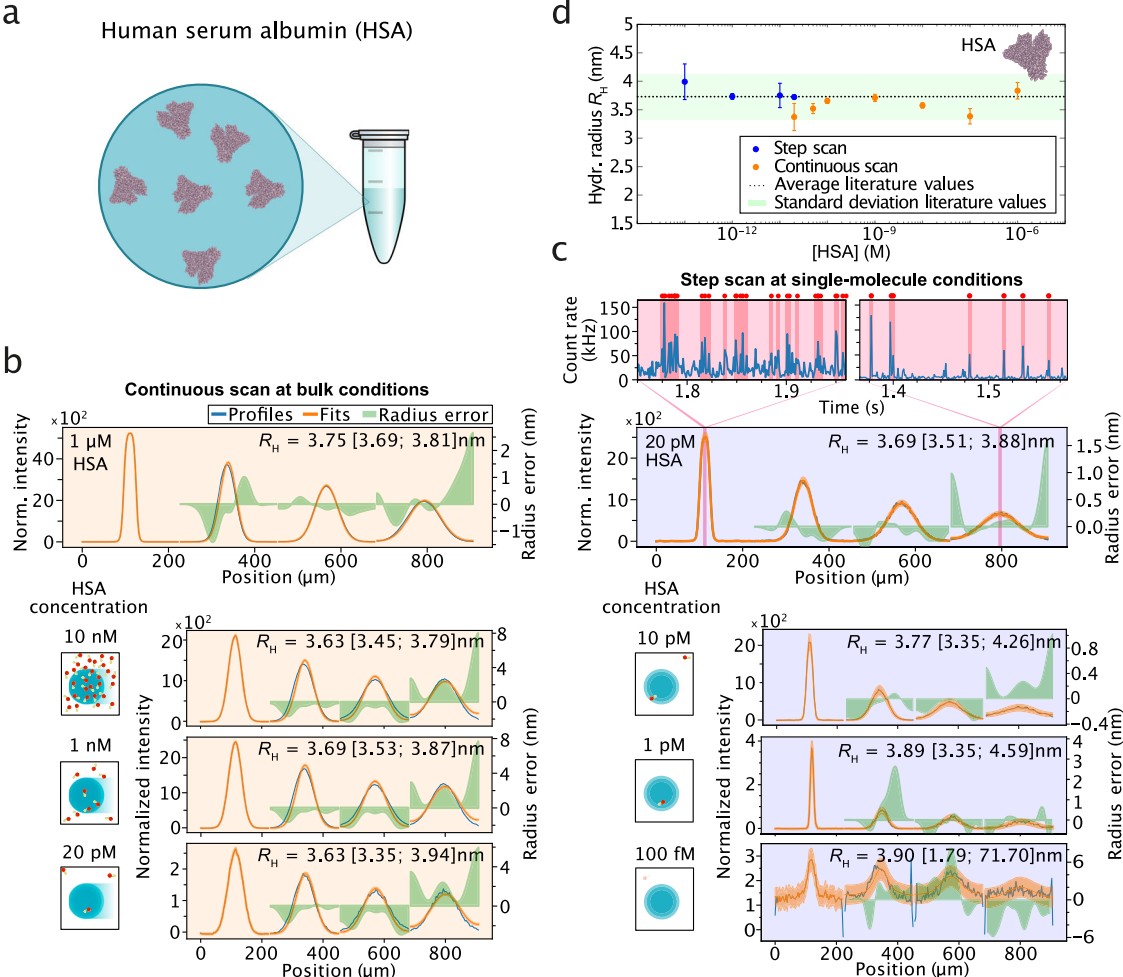

**Fig. 2 | Sizing of proteins from bulk to single-molecule conditions by smMDS.**
**a** The sensitivity of smMDS and its capability to size proteins from bulk to single-molecule conditions was evaluated by measuring the size of human serum albumin (HSA) at varying protein concentrations. **b** Diffusion profiles for HSA as obtained from continuous scan measurements. From top to bottom: 1 µM, 10 nM, 1 nM, 20 pM HSA. Diffusion profiles are shown as blue lines, experimental fits as orange lines, and errors as green bands. Extracted hydrodynamic radii $R_H$ [with errors] are given as insets. For definitions of errors, please refer to the legend of Fig. 1b. Schematics on the left depict the decrease in concentration. **c** Diffusion profiles for HSA obtained from step scan measurements under single-molecule conditions. From top to bottom: 20 pM, 10 pM, 1 pM, 100 fM HSA. Diffusion profiles are shown as blue lines, experimental fits as orange lines, and errors as green bands. Extracted

hydrodynamic radii $R_H$ [with errors] are given as insets. For definitions of errors, please refer to the legend of Fig. 1b. Schematics on the left depict the decrease in concentration. The two highlighted plots on the top are exemplary single-molecule time trajectories recorded at two channel positions, as indicated. Red dots and highlighting indicate bursts detected by the burst-search algorithm. **d** $R_H$ of HSA as obtained by continuous scan (orange points) and step scan (blue points) measurements. Data points (mean) were obtained from at least triplicate measurements at the respective sample concentration (see source data for number of repeats). Error bars denote standard deviations. The dashed line indicates the average literature value (mean) for HSA ($R_H = 3.73 \pm 0.40$ nm)[67–70] with the green band depicting the standard deviation of literature values. Source data are provided as a Source Data file.

range of diffusional sizing experiments by more than five orders of magnitude.

## Sizing of proteins and protein assemblies by smMDS

Next, we sought to demonstrate the wide applicability of smMDS in determining the size of proteins and protein assemblies from single-molecule digital counting. We selected a varied set of analytes differing in size, including the proteins lysozyme, RNase A, α-synuclein, human leukocyte antigen (HLA), HSA, thyroglobulin, and oligomers formed by the protein α-synuclein. These protein analytes collectively span a size range of 1–10 nm. We also included the small organic fluorophore Alexa 488 in the series as a sub-nm sized analyte (Fig. 3a). The protein analytes were fluorescently labeled and purified before analysis (see Methods and Supplementary Table 1). We performed smMDS measurements at an analyte concentration of 10 pM and subjected the analytes to smMDS in step scan mode. We moved the confocal spot in a stepwise manner through the channels

and extracted single-molecule events for each analyte at each channel position by digitally counting molecules to create diffusion profiles, which we fitted with our advection–diffusion model. Exemplary diffusion profiles for thyroglobulin, HLA, and Alexa 488 are shown in Fig. 3b. The obtained $R_H$ values from smMDS were then plotted against previously reported $R_H$ values for Alexa 488[72], lysozyme[73], RNase A[74], α-synuclein[75], human leukocyte antigen (HLA)[53], HSA[67–70], thyroglobulin[76], and α-synuclein oligomers[77] (Fig. 3a). The values obtained by smMDS followed the expected trend within error. This demonstrates the excellent agreement between sizes obtained from smMDS and literature values, highlighting the reliability of the single-molecule diffusivity measurements in size determination of protein analytes. In an additional analysis step, we plotted the experimentally obtained $R_H$ values against the molecular weight $M_W$. Both, folded proteins (lysozyme, RNase A, HLA, HSA, thyroglobulin) and unfolded protein species (α-synuclein monomer and oligomers), followed the expected scaling

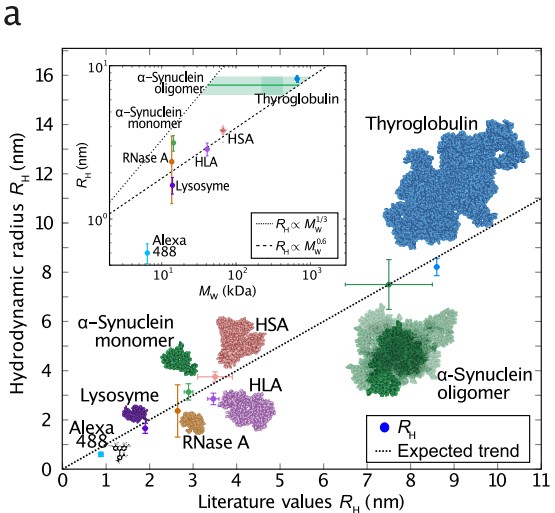

a

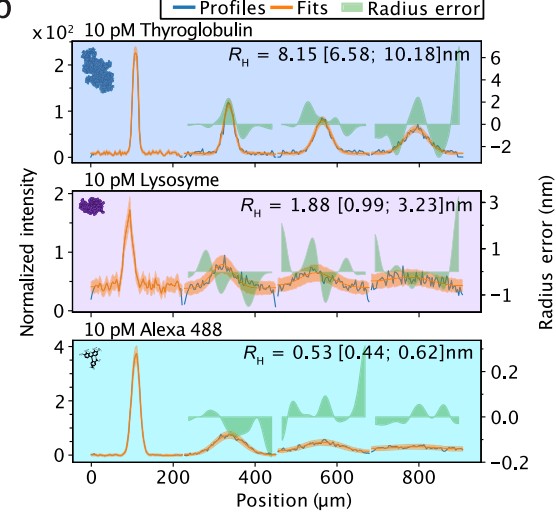

b

**Fig. 3 | Sizing of proteins and protein assemblies by smMDS. a** Experimentally determined hydrodynamic radii $R_H$ for various protein species as obtained from smMDS plotted against literature values. The dashed line depicts the expected trend. Data points (mean) were obtained from at least triplicate measurements at 10 pM sample concentration (see source data for a number of repeats). Error bars denote standard deviations. Inset shows obtained $R_H$ (mean ± standard deviation) as a function of molecular weight $M_W$. The dashed and dotted lines denote scaling

behavior of globular ($R_H \propto M_W^{1/3}$) and disordered ($R_H \propto M_W^{0.6}$) proteins, respectively. **b** Exemplary diffusion profiles for thyroglobulin (blue), human leukocyte antigen (HLA) (violet), and Alexa 488 (cyan) at 10 pM sample concentration. Diffusion profiles are shown as blue lines, experimental fits as orange lines, and error as green bands. Extracted $R_H$ [with errors] are given as insets. For definitions of errors, please refer to the legend of Fig. 1b. Source data are provided as a Source Data file.

behavior for globular ($R_H \propto M_W^{1/3}$) and disordered ($R_H \propto M_W^{0.6}$) proteins, respectively (Fig. 3a, inset).

## Quantifying protein interactions by smMDS

Next, we set out to demonstrate the capability of smMDS in determining the affinity of biomolecular interactions at the single-molecule level. Interactions of proteins with secondary biomolecules, in particular with other proteins, are of great importance across the biosciences, and quantitative measurements of affinity constants in the form of $K_D$s have become vital in biomedical research and clinical diagnostics, for example, for histocompatibility testing and affinity profiling[46,53,54,78]. Diffusional sizing allows for the detection of biomolecular interactions by monitoring the increase in size associated with binding and complex formation[42,48]. By acquiring binding isotherms, affinity constants of the interaction can be determined in solution, without the need for purification or for immobilization on a surface. So far, diffusional sizing has been limited to the sizing and quantification of protein interactions at bulk nanomolar concentration levels—with smMDS, this barrier can be overcome.

To demonstrate the detection of biomolecular interactions and quantification of binding affinities by smMDS in a digital manner, we probed the binding of a clinically relevant antibody–antigen interaction. Specifically, we investigated the binding interaction between HLA A*03:01, an isoform of the major histocompatibility complex type I (MHC) and a key factor in the human immune system[79], and the antibody W6/32, an antibody that binds to all class I HLA molecules (Fig. 4a)[80]. We performed a series of step scan smMDS experiments by titrating HLA antigen (labeled with Alexa 488), at a constant concentration of 100 pM, with increasing amounts of the unlabeled W6/32 antibody. We opted for labeled HLA and unlabeled W6/32 antibody in our study, motivated by the clinical significance of detecting anti-HLA antibodies in patient serum, especially in scenarios involving organ transplantation[53]. Exemplary diffusion profiles for pure HLA at 100 pM, and 100 pM HLA titrated with 10 nM of W6/32 are shown in Fig. 4b. smMDS diffusion profiles, from three repeats, were acquired and fitted to obtain effective $R_H$ across the concentration series. We observed an increase in average hydrodynamic radius from $R_H = 3.18 \pm 0.04$ nm for

pure HLA, corresponding to a molecular weight of 50 kDa, as expected for HLA, to $R_H = 5.08 \pm 0.01$ nm for the saturated complex, corresponding to a molecular weight of 215 kDa, consistent with the binding of a 150 kDa antibody to HLA (Fig. 4c). By fitting the binding isotherm (Fig. 4d), we determined the dissociation constant to be $K_d = 400.5 \pm 39.6$ pM, consistent with previous results[53]. Importantly, HLA antibodies are an extensively used clinical biomarker to evaluate, for example, histocompatibility and the risk of allograft rejection[53]. Given that these antibodies are usually found in patient serum at very low concentrations, our findings offer a method for detecting and profiling antibody responses even when only minimal sample quantities are available.

Overall, our results here highlight the potential of quantifying biomolecular interactions through single-molecule digital measurements, even at very low concentrations. We note that the possibility to measure at very low protein concentrations enables examining high-affinity interactions (i.e., with sub-nanomolar affinities). In these cases, the binding curves are shifted significantly to the left, necessitating an approach capable of resolving binding curves at lower concentration ranges. Traditional bulk assays typically lack the sensitivity for such low concentrations. Step scan smMDS, by contrast, could provide this resolution to delineate high-affinity interactions. Furthermore, it is noteworthy that the single-molecule binding assay demonstrated here could be applied to other antibody–antigen systems (e.g., profiling of SARS-CoV-2 antibody interactions)[46,50,54] and the labeling scheme, with labeled antigen and an unlabeled antibody, could be customized to suit the specific diagnostic requirements of the system being investigated. For example, if the detection of low concentrations of antigen is more relevant in a certain diagnostic context, the roles can be reversed, with the antibody being labeled and the antigen remaining unlabeled[42,48,81].

## Resolving protein oligomeric states by smMDS

Many proteins fulfill their biological roles not as monomeric species but as oligomeric assemblies, which often exhibit significant heterogeneity in terms of their degree of oligomerization and relative abundance[82–84]. Oligomeric forms of proteins play important functional roles in cellular physiology but are also implicated in diseases

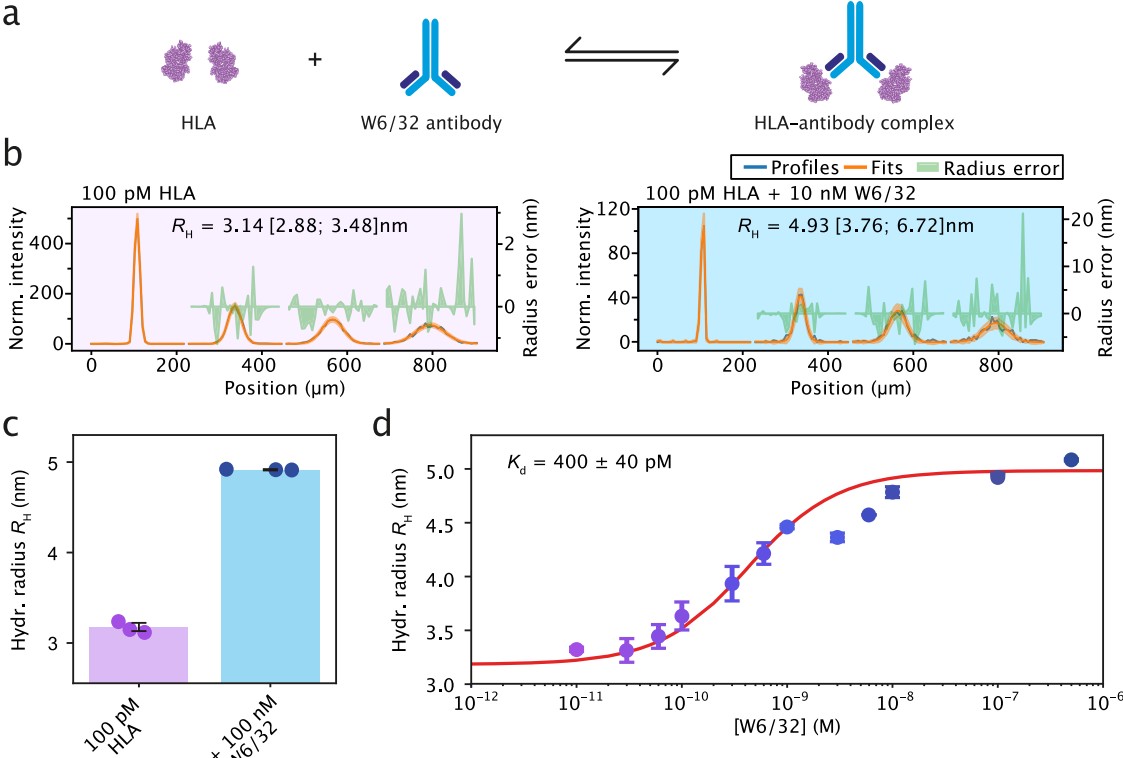

**Fig. 4 | Quantifying protein interactions by smMDS. a** Sizing and affinity measurement of an antibody–antigen complex by smMDS. Shown is a schematic for the binding interaction between human leukocyte antigen (HLA) (A*03:01) and the HLA-antibody W6/32. **b** Exemplary diffusion profiles for the binding of the HLA-antibody W6/32 to HLA as obtained by step scan smMDS. The left panel shows a diffusion profile for 100 pM HLA. The right panel shows a diffusion profile for 100 pM HLA in the presence of 10 nM W6/32. Diffusion profiles are shown as blue lines, experimental fits as orange lines, and error as green bands. Extracted hydrodynamic radii ($R_H$) [with errors] are given as insets. For definitions of errors, please refer to the legend of Fig. 1b. **c** Size increase upon complexation of HLA with W6/32

from $R_H = 3.18 \pm 0.04$ nm for pure HLA to $R_H = 5.08 \pm 0.01$ nm for the complex in the presence of 100 nM W6/32. Data points (mean) were obtained from triplicate measurements. Error bars denote standard deviations. Notably, for enhanced clarity and to prevent overlap, data points are randomly positioned along the x-axis in each chart. **d** Binding isotherm obtained from a titration of 100 pM HLA with increasing concentrations of W6/32. For analysis, the binding isotherm was fitted with a binding model assuming two antigen molecules binding one antibody[53]. The dissociation constant was found to be $K_d = 400 \pm 40$ pM. Error bars are standard deviations from triplicate measurements. Source data are provided as a Source Data file.

such as neurodegeneration[85–87]. Resolving the degree of oligomerization and thus the size of such heterogenous protein populations is, however, challenging with currently available biophysical techniques. A key feature of smMDS is that it has the capability to directly distinguish between various assembly states of a protein based on a difference in their emitted fluorescence signals. This feature is afforded by the single-molecule sensitivity of smMDS and enables the creation of diffusion profiles from subspecies that make up the heterogeneous population. To demonstrate this capability, we set out experiments with two protein oligomer systems that are inherently heterogeneous and have distinct functions in biology and disease.

In a first set of experiments, we determined the sizes of low-molecular weight oligomers formed by the protein HSA (Fig. 5). Serum albumins are known to exist in an equilibrium of monomers, dimers, trimers, and tetramers, which are populated with decreasing relative abundance[32,71]. At the single-molecule level, such different oligomeric states can be discriminated through brightness analysis of fluorescence bursts[24,25]. In this analysis, different species can be distinguished based on their emitted fluorescence intensity because the magnitude of the observed intensity scales directly with the number of individually dye-labeled monomer units present in an oligomeric assembly. By applying differential thresholding, oligomeric states can then be discriminated, which provides an opportunity for smMDS to create distinct diffusion profiles for each oligomeric subspecies, enabling their independent species-specific sizing. To demonstrate this, we set out smMDS measurements to resolve the sizes of HSA monomers, dimers,

trimers, and tetramers. We subjected labeled HSA to smMDS at 10 pM protein concentration and performed step scan measurements to extract single-molecule events from single-molecule time trace recordings. We then displayed the extracted normalized burst intensities from all recorded burst events of the measurement in a burst intensity histogram (Fig. 5a). This allowed us to display single-molecule burst events according to their brightness and assign regions of intensity for the monomeric and the different oligomeric HSA species. Accordingly, the main peak in the histogram reflects the average intensity of monomeric HSA. We extracted this intensity by fitting the distribution with a skew-normal distribution function, which reflects the skewedness of the burst intensity distribution due to the under-sampling effects at short burst times, and retrieved a mean intensity for the monomer of $I_{monomer} = 75.33$ photons/ms with a standard deviation $\sigma_{monomer}$ of 37.44 photons/ms. Since oligomers contain as many fluorophores as monomer units, dimeric, trimeric, and tetrameric forms of HSA emit at multiples of the normalized intensity of monomeric HSA, due to the increasing number of fluorophore present within the assembled states. We therefore defined regions at two-, three-, and four-fold of the normalized intensity of the monomer, corresponding to HSA dimer, trimer, and tetramer, respectively by fitting the burst intensity distribution with three Gaussian functions. The widths of these oligomer regions were assumed to have the same standard deviation as the monomeric protein. The resulting fit for all Gaussians, including the skew-normal distribution for the monomer, described well the experimental burst intensity distribution. Notably,

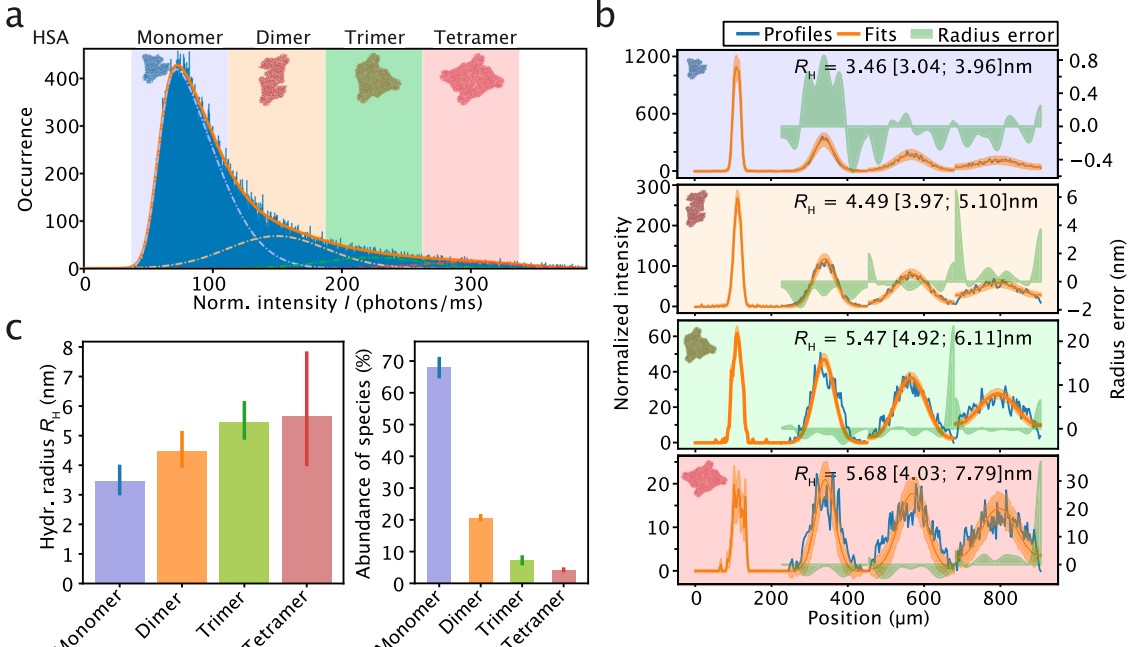

**Fig. 5 | Resolving protein oligomeric states by smMDS. a** Sizing of low-molecular weight oligomers formed by the protein human serum albumin (HSA). Shown is a burst intensity histogram, which displays all bursts extracted from a single smMDS step scan measurement of HSA (10 pM). Intensities are normalized intensities with respect to burst duration. Four regions (blue, orange, green, and red), which correspond to HSA monomer and HSA dimer, trimer, and tetramer, are defined in the burst intensity histogram. These were obtained by fitting the distribution with a skew-normal distribution function for monomeric HSA and three Gaussian functions for dimeric, trimeric, and tetrameric HSA (see main text for details). The center positions for the oligomers (dimer: 150.66 photons/ms, trimer: 225.99 photons/ms, and tetramer: 301.32 photons/ms) are multiples of the normalized intensity of the monomer ($I_{monomer}$ = 75.33 photons/ms). The widths of the regions reflect one standard deviation of the distribution of monomeric HSA ($\sigma_{monomer}$ = 37.44 photons/ms). **b** Diffusion profiles for HSA monomer, dimer,

trimer and tetramer generated from bursts within each of the four regions in the burst intensity histogram shown in panel a (panels are color-coded according to the colors used in panel a). Each profile was fitted to extract size information. Diffusion profiles are shown as blue lines, experimental fits as orange lines, and error as green bands. Extracted hydrodynamic radii $R_H$ [with errors] are given as insets. For definitions of errors, please refer to the legend of Fig. 1b. **c** Species-resolved $R_H$ (left panel) and abundance of HSA oligomers (right panel). $R_H$ were obtained from diffusion profile fits shown in panel b and represent the best-fit values; error bars correspond to the error range of $R_H$ as derived from the global fit. For definitions of errors, please refer to the legend of Fig. 1b. Abundance was obtained from skew normal/Gaussian fitting in panel a and represents the best-fit value; the error bars correspond to the 99% confidence intervals obtained from bootstrapping. Source data are provided as a Source Data file.

the four-species model is validated by statistical analysis, as shown in Supplementary Fig. 3, indicating that a four-species model aligns best with the data (see Supplementary Note 2 for more information). We then generated diffusion profiles from the bursts within each of the four regions and fitted the profiles to extract size information from the respective monomer/oligomer range (Fig. 5b). The extracted sizes of the four different regions correspond to molecular weights of proteins of 65 (44–99) kDa, 145 (99–214) kDa, 266 (192–370) kDa, and 298 (192–790) kDa, respectively, and thus are, within error, in very good agreement with the sizes of monomeric, dimeric, trimeric, and tetrameric HSA (66 kDa, 144 kDa, 199 kDa, 266 kDa, respectively). In addition to subspecies-resolved sizing of oligomers, our analysis can also provide information on the relative abundance of oligomeric species. The areas under the curves, as obtained from Gaussian fitting of the burst intensity histogram, reflect the abundance of oligomeric species. We obtained relative abundances of 67.9% for the monomer, 20.7% for the dimer, 7.2% for the trimer, 4.2% for the tetramer (Fig. 5c). These values are in broad agreement with the equilibrium distribution of other serum albumins. For example, for bovine serum albumin, relative abundances of 88.63%, 9.94%, 1.18% and 0.25% for monomers, dimers, trimers, and tetramers were previously found[32]. Overall, our analysis here shows that smMDS can afford species-resolved insights into oligomer size and abundance.

In a further set of experiments, we analyzed a heterogenous mixture of α-synuclein oligomers (Fig. 6). Oligomeric forms of the protein α-synuclein are considered to be central to the pathology of

Parkinson's disease and hallmarked by a high degree of heterogeneity in terms of size and structure[88–90]. Their characterization is an area of intense interest, not least because such information is useful in drug development activities, however, tools that can directly resolve the heterogeneity of these nanoscale assemblies in solution are scarce[77,91–95]. To address this challenge and characterize the structural heterogeneity of α-synuclein oligomers, we analyzed a heterogenous mixture of α-synuclein oligomers by smMDS. We injected oligomers produced by lyophilization of Alexa 488-labeled α-synuclein into the microfluidic sizing chip at a concentration of 10 pM and performed step scan measurements to digitally extract single-molecule events of passing α-synuclein oligomer molecules at each channel position. To create diffusion profiles from subspecies, we selected bursts with different fluorescence intensities to resolve differently sized assembly states of oligomers within the mixture. Here we took an alternative approach as compared to the analysis of HSA oligomers presented above and extracted bursts by varying the minimum number of fluorescence photons in the burst search algorithm, while keeping the inter-photon time threshold constant. This allowed us to effectively differentiate between single-molecule burst events that differ in their molecular brightness (see burst intensity histograms shown in Fig. 6a). In this way, diffusion profiles from assemblies that differ in their fluorescence intensity and, hence, size in terms of $R_H$ were generated. Exemplary diffusion profiles for four different thresholds are shown in Fig. 6b. These profiles were then fitted with our advection–diffusion model to extract size information. We applied this analysis to a range

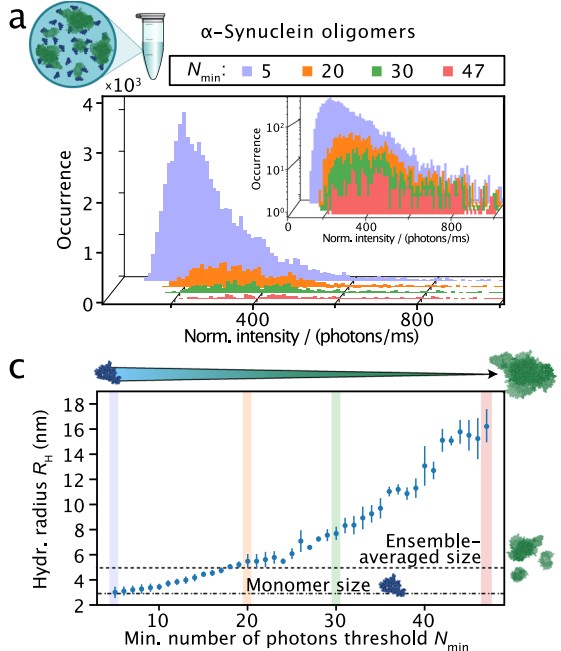

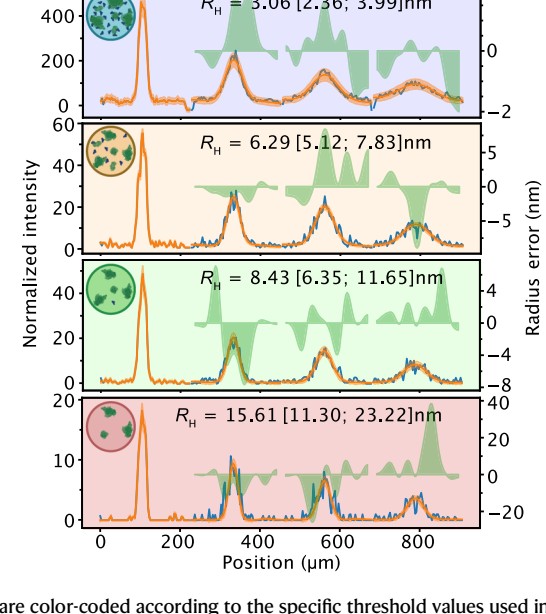

**Fig. 6 | Sizing of heterogenous oligomer populations by smMDS. a** A heterogenous mixture of α-synuclein oligomers (10 pM) was probed in a single-step scan smMDS measurement and single-molecule burst events were extracted using the burst search algorithm. To differentiate between differently sized assembly states of α-synuclein oligomers, the value for the minimum number of fluorescence photons threshold ($N_{min}$) was varied, while keeping the inter-photon time threshold constant. This allowed for the creation of burst intensity distributions, which differ in molecular brightness. Exemplary burst intensity distributions for four different $N_{min}$ threshold values are shown (light blue: 5 photons, orange: 20 photons, green: 30 photons, red: 47 photons). The inset displays burst intensity histograms in semi-log scale. Intensities are normalized intensities with respect to burst duration. **b** Exemplary diffusion profiles generated from burst intensity distributions with four different minimum number of fluorescence photons threshold values (panels are color-coded according to the specific threshold values used in panel a). Diffusion profiles are shown as blue lines, experimental fits as orange lines, and error as green bands. Extracted hydrodynamic radii $R_H$ [with errors] are given as insets. For definitions of errors, please refer to the legend of Fig. 1b. **c** Extracted $R_H$ of oligomer subspecies displayed as a function of the different minimum number of photon threshold values used in the burst search algorithm. Data represent extracted $R_H$ [with errors], as reported in panel b. The colored vertical bars indicate threshold values used in panels a and b. The horizonal dashed line indicates the ensemble-averaged size of the entire oligomer population, generated from a diffusion profile from all bursts obtained from the measurement. $R_H$ of monomeric α-synuclein, as measured by smMDS, is indicated as a dotted horizontal line. Source data are provided as a Source Data file.

of photon thresholds, specifically between 5 and 50 total photons, to ensure that the chosen thresholds were representative of the intensity variations observed in oligomer burst signals. We then generated a plot of the extracted sizes versus photon thresholds from the diffusion profile series (Fig. 6c). The measured sizes span from $R_H = 3.6$ nm for the smallest photon threshold value to $R_H = 16.5$ nm for the largest threshold value. The value at the smallest threshold reflects the size of α-synuclein monomers ($R_H = 3.1 \pm 0.4$ nm) (c.f. Figure 3), while higher values reflect α-synuclein oligomer subpopulations, spanning a range of $R_H$ values greater than 10 nm. The $R_H$-value distribution obtained by smMDS is in very good agreement with the size ranges reported in earlier studies, as determined by techniques such as atomic force microscopy (AFM)[91,96–98], transmission electron microscopy (TEM)[99–103], small-angle X-ray scattering (SAXS)[92,99,104–106], and dynamic light scattering (DLS)[104,107]. In comparison, we also generated a diffusion profile from the entire set of bursts detected at each position, which through fitting yielded an ensemble-averaged value of the size of the oligomer population ($R_H = 5.2 \pm 0.1$ nm) (Fig. 6c). To complement the analysis demonstrated here based on varying the minimum number of fluorescence photon thresholds in the burst search algorithm, we also performed size analysis of α-synuclein oligomers by selecting defined regions in the burst intensity histogram as done for HSA. The results are shown in Supplementary Fig. 4 and are in very good agreement with the results obtained by varying the fluorescence photon thresholds value (c.f. Figure 6c). Taken together, our analyses here demonstrate the versatility of smMDS in resolving the size distributions of a heterogenous oligomeric protein samples.

## Sizing of multiple species within a heterogenous aggregation mixture by smMDS

Many protein systems are heterogeneous, multicomponent mixtures consisting of proteins and protein assemblies that differ in size by several orders of magnitude. For example, aggregation mixtures are made up of monomeric protein and large fibrillar species[3,47]. Often, one of the components (e.g., the monomeric protein) is present in large excess, while the other one (e.g., fibrillar species) is only present in small amounts. Approaches that can quantify the sizes of such differently populated species are much sought after yet lacking. smMDS can fill this gap as it has the capability to size molecules and assemblies in heterogeneous mixtures even when an excess of one of the molecular species is present at bulk levels.

To demonstrate the potential of sizing protein mixtures that are compositionally heterogeneous, we set out experiments with a sample system composed of fibrils formed by the protein α-synuclein, a key component in the pathology of Parkinson's disease[108,109], and an excess of monomeric α-synuclein at nanomolar concentrations (Fig. 7a). Such a mixture is often encountered in assays that probe the mechanisms underlying protein aggregation and amyloid formation[47,110]. We first performed continuous scanning experiments on pure α-synuclein fibrils (at 10 nM monomer equivalent concentration) and pure α-synuclein monomer (at 10 nM concentration) to establish the signature of the two species (Fig. 7b). Notably, within the fibril sample, only 10% of the monomers are fluorescently labeled (see Methods). Similarly, only a fraction of 10% of labeled protein is present in the monomer sample, thus ensuring

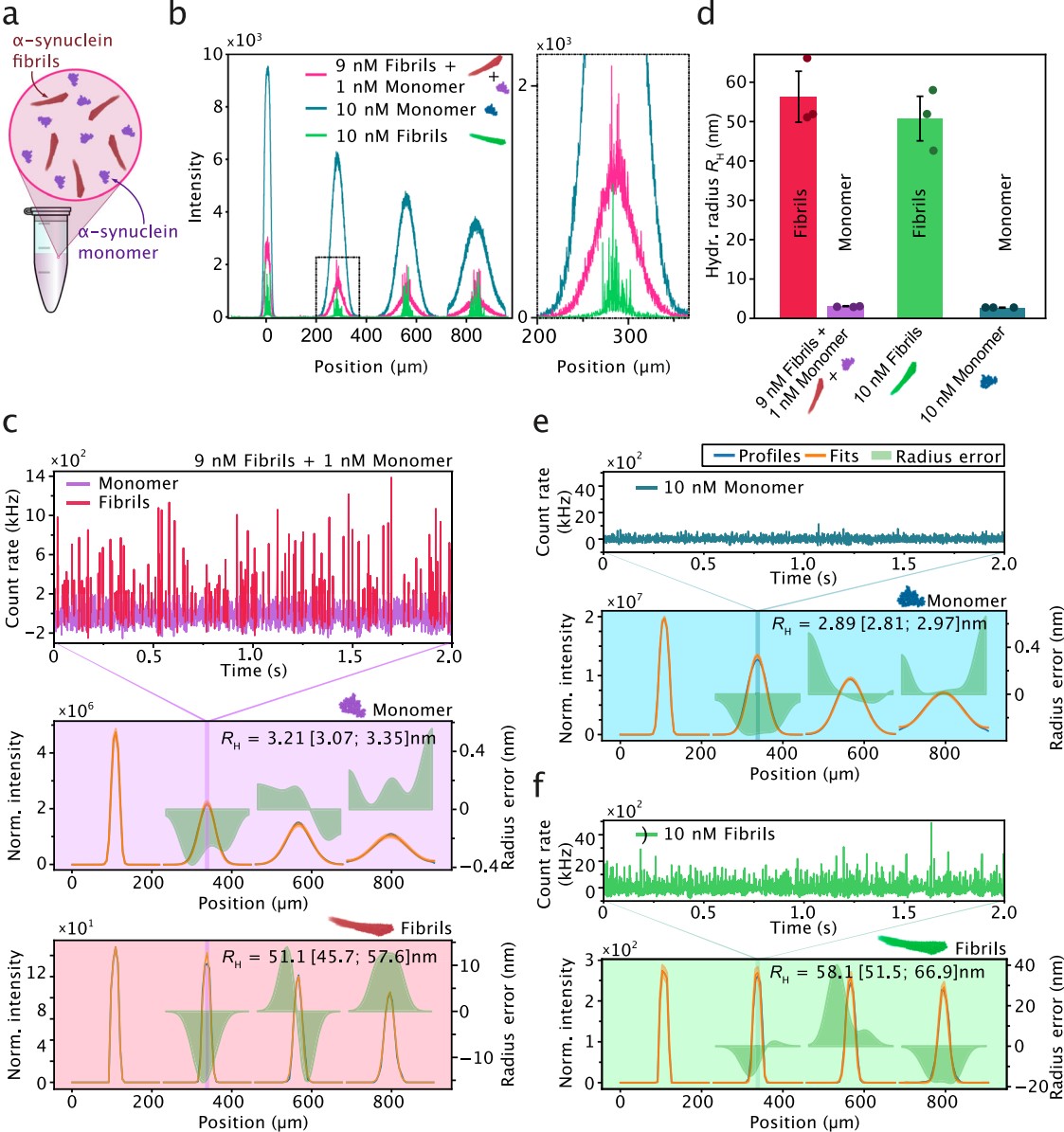

**Fig. 7 | Sizing of multiple species within a heterogenous aggregation mixture by smMDS. a** Schematic of an aggregation reaction composed of monomeric α-synuclein and fibrillar species. **b** Sizing of α-synuclein fibrils in the presence of an excess of monomeric α-synuclein. Continuous scan diffusion profiles (left panel) for pure monomeric α-synuclein (10 nM) (blue), α-synuclein fibrils (10 nM monomer equivalent) (green), and a mixture of α-synuclein fibrils (9 nM monomer equivalent) and monomeric α-synuclein (1 nM) (pink). The right panel is a zoom-in as indicated by a dashed box in the left panel. Bursts correspond to the passing of fibrils through the confocal detection volume. **c** Step scan measurement of a mixture of α-synuclein fibrils (9 nM monomer equivalent) and monomeric α-synuclein (1 nM). The top panel shows an exemplary fluorescence time trace (1-ms binning) at diffusion profile position 340 μm, as indicated. An intensity threshold was applied to separate signal from fibrils (red) and monomer (purple). The bottom panels show diffusion profiles created from the fibril and monomer signals, respectively. Diffusion profiles are shown as blue lines, experimental fits as orange lines, and errors as green bands. Extracted hydrodynamic radii $R_H$ [with errors] are given as insets. For definitions of errors, please refer to the legend of Fig. 1b. **d** Comparison of extracted sizes from triplicate step scan measurements. Shown are $R_H$ of species extracted from a mixture of α-synuclein fibrils (9 nM monomer equivalent) and 1 nM monomeric α-synuclein (red and purple, respectively), pure monomeric α-synuclein (blue), 10 nM monomer equivalent of α-synuclein fibrils (green). Data points (mean) were obtained from triplicate measurements. Error bars denote standard deviations. Step scan measurements of pure α-synuclein (10 nM) (panel **e**) and pure fibrils (10 nM monomer equivalent) (panel **f**). The top panels show exemplary fluorescence time traces (1-ms binning) at diffusion profile positions 338 μm and 340 μm, respectively. Diffusion profiles are shown as blue lines, experimental fits as orange lines, and errors as green bands. Extracted $R_H$ [with errors] are given as insets. For definitions of errors, please refer to the legend of Fig. 1b. Source data are provided as a Source Data file.

concentration parity with the fibrils and facilitating direct comparisons between the two.

For the fibril-only sample (Fig. 7b, green profile), we observed burst events of high fluorescence intensity that were narrowly distributed around the center of the channels. The high burst intensity stems from the large number of fluorophores that are contained in a single fibril (>10% of the monomers are fluorescently labelled within

fibrils, see Methods). The narrow distribution of bursts located at the center of the channel indicates a low diffusion coefficient and correspondingly a large size, as expected for fibrillar aggregates. For the monomer sample (Fig. 7b, blue profile), the sizing profiles exhibited a wider spread. This is attributed to the higher diffusivity of monomers compared to fibrils. The monomer signal is continuous because nanomolar concentrations are used, and therefore

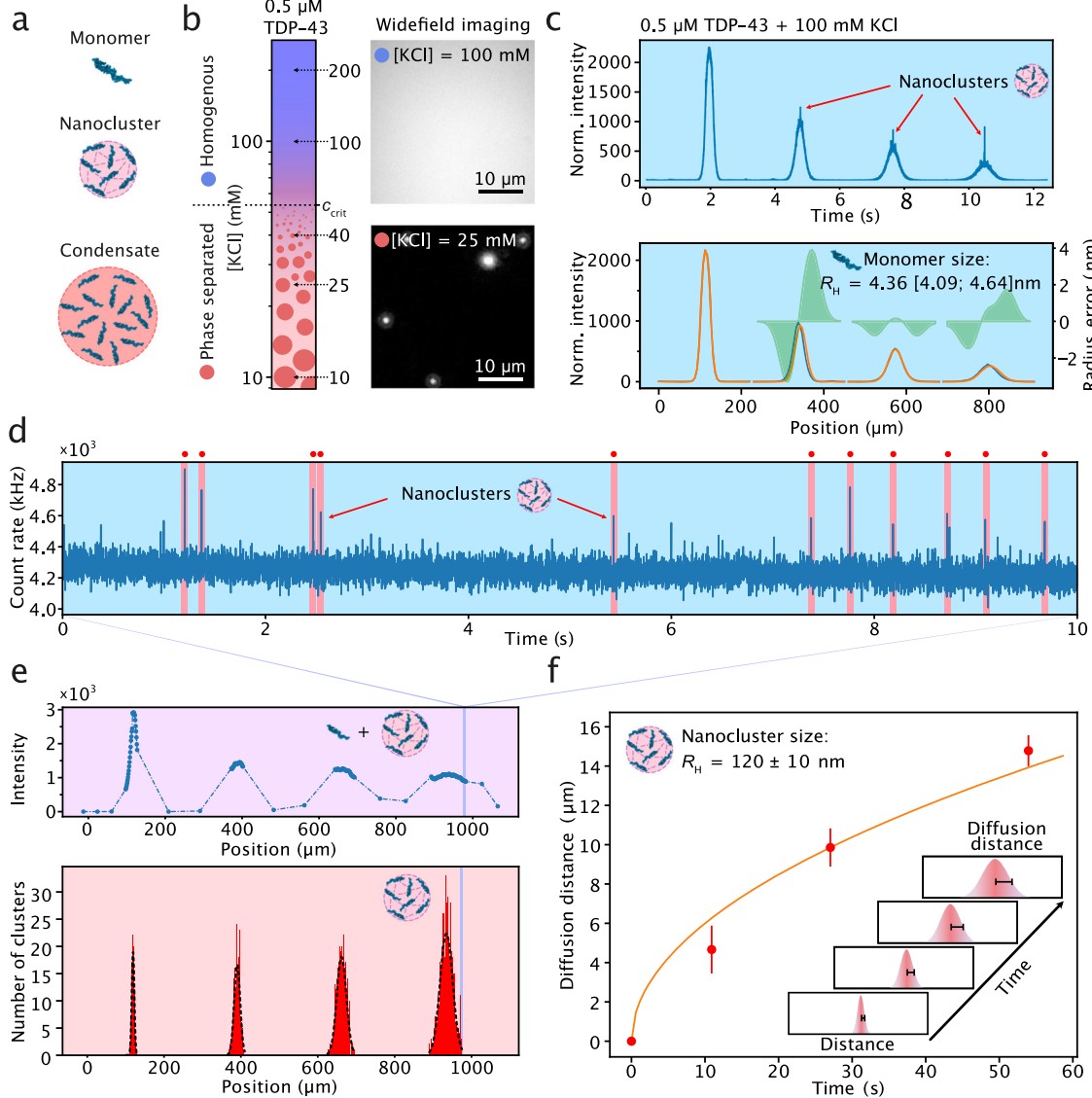

**Fig. 8 | Sizing of nanoscale clusters by smMDS. a** Schematic of TDP-43 phase separation and the formation of nanoscale clusters in the pre-phase separating regime. **b** Phase separation behavior of GFP-tagged TDP-43 as a function of KCl concentration as observed by widefield microscopy imaging. The phase diagram (left panel) was generated from measurements at five KCl concentrations and at 0.5 μM protein concentration. Representative images at 100 mM and 25 mM KCl are shown (right panels). $c_{crit}$ denotes the critical KCl concentration. Experiments were repeated at least three times with similar results. Scale bar is 10 μm. **c** Continuous scan diffusion profiles for 0.5 μM GFP-tagged TDP-43 at 100 mM KCl. The upper panel shows the diffusion profile as obtained from a continuous scan measurement. Bright bursts indicate nanoclusters passing through the confocal detection volume. The bottom panel is a re-binned diffusion profile to extract the size of monomeric TDP-43. Diffusion profiles are shown as blue lines, experimental fits as orange lines, and error as green bands. The extracted $R_H$ [with errors] is given as an inset.

**d** Exemplary fluorescence time trace (1-ms binning) from a step scan measurement at channel position 960 μm, as indicated in panel e. Nanoclusters were detected as bursts that exhibit a signal >5 standard deviations above the mean. Detection events are highlighted in red. **e** Total intensity of a segmented step scan across the chip (top panel) and histogram of detected nanocluster events as a function of chip position (bottom panel). Gaussian distributions were fit to each peak to extract a mean diffusion distance at each channel position. **f** Plot of mean diffusion distance versus time of travel within the channel. The inset graphically shows how diffusion distances were determined. The diffusion distance corresponds to the half of the full-width half maximum (FWHM) of the Gaussian distributions at each measurement point. The width at timepoint zero was used for normalization. Data points (mean) are from three repeats; error bars indicate standard deviations. The orange line shows the fit according to Eq. 1. The extracted average $R_H$ of TDP-43 nanoclusters is given as an inset (mean ± standard deviation).

multiple monomeric units traverse the confocal detection volume at the same time, resulting in a bulk fluorescence signal contrasting with the discrete single-molecule events observed in the fibril sample. In addition to establishing the signatures of fibrillar and monomeric samples, we probed a sample mix containing α-synuclein fibrils and an excess of the monomeric protein (Fig. 7b, pink profile). The diffusion profile of the mixture exhibited characteristic signatures for both fibrils and monomeric protein, with broadened fluorescence at the profile base, reflecting monomeric protein, in addition to bright bursts on top of the monomeric signal

that were narrowly distributed at the center of the channel, reflecting signals from fibrils.

To demonstrate that smMDS is able to size both the monomeric subpopulation present at bulk levels and the fibrils present at single-particle concentrations, we performed step scan measurements with a mix containing α-synuclein fibrils and an excess of the monomeric protein (Fig. 7c). An example fluorescence time trace is shown in Fig. 7c (top panel). Fibrils are clearly detectable as bursts above the mean signal, which corresponds to the bulk monomer signal. From these traces, we separated the bulk monomer signal from the fibril burst

signals by intensity thresholding. Specifically, fibrils were detected as bursts that exhibit a fluorescence count rate of >250 kHz, after applying a Savitzky-Golay smoothing filter. The remaining signal (i.e., the mean bulk signal in the fluorescence time traces in the absence of fibril signal) formed the signal for the monomer. From the extracted fibril and monomer signal, we created diffusion profiles for the two species (Fig. 7c, bottom panels) and subjected these profiles to fitting using our advection–diffusion model to extract size information of the two species. The sizes of monomer and fibrils species, from triplicate measurements (Fig. 7d), were estimated to be $R_{H,monomer} = 3.23 \pm 0.04$ nm and $R_{H,fibrils} = 56.43 \pm 6.69$ nm. As a control, we also performed step scan measurements for the fibril-only and monomer-only sample for comparison (Fig. 7e, f, respectively) and obtained sizes which were, within error, in excellent agreement with the ones obtained from the sample mix, thereby validating our approach (Fig. 7d).

Together, these experiments here show that smMDS has the capability to quantify differently sized molecules or assembly states of a protein within a heterogeneous mixture, even when an excess of one of the molecular species is present at bulk levels. These findings are significant as such an approach allows for the simultaneous probing of differently populated species, for example, in kinetic protein misfolding and aggregation studies.

## Sizing of nanoscale clusters by smMDS

In a final set of experiments, we applied the smMDS approach to the characterization of nanometer-sized clusters of a phase separating protein system. Biomolecular condensates (Fig. 8a) formed through phase separation are important players in cellular physiology and disease[111,112], and emerge from the demixing of a solution into a condensed, dense phase and a well-mixed, dilute phase[113,114]. Condensates typically have sizes in the micrometer range and are easily observable by conventional microscopy imaging[115–117]. Recent evidence suggests that phase separation-prone proteins, such as the DNA/RNA binding protein fused in sarcoma (FUS), can also form nanoscale assemblies (Fig. 8a), well-below the critical concentration at which phase separation occurs (i.e., pre-phase separating regime)[118–121]. These so-called nanoscale clusters have sizes in the tens to hundreds nanometer regime, and thus are beyond the resolution of conventional optical imaging systems, meaning that their precise quantification of the cluster dimensions is infeasible. Moreover, as these species are low in abundance and present in a high background of dilute phase protein concentration, they are typically hard to detect by conventional wide-field imaging approaches. This is because wide-field fluorescence microscopy inherently captures significant out-of-focus light from the entire sample volume. Due to the high concentration of protein in the dilute phase, this leads to a high background signal that effectively masked the distinct fluorescence signal emanating from the clusters. Consequently, these clusters, while theoretically visible, become indistinguishable from the overwhelming background noise in standard wide-field epifluorescence images. This issue can be circumvented with confocal fluorescence detection, which utilizes optical sectioning to restrict detection to the in-focus plane only. Here, we showcase how smMDS, leveraging confocal detection to achieve nanocluster detection sensitivity, enables sizing of nanoscale assemblies of TAR DNA binding protein 43 (TDP-43) at sub-saturating concentrations directly in solution. Additionally, we illustrate smMDS' efficacy in assessing the abundance and composition of these nanoclusters.

First, we mapped out a one-dimensional phase diagram of the protein TDP-43 (Fig. 8b) to assess the phase separation behavior of the protein with respect to changes in salt concentration. GFP-tagged TDP-43 at 0.5 μM protein concentration formed microscopically visible condensates (1–2 μm in diameter) below a critical salt concentration $c_{crit}$ of 50 mM KCl. No condensates were visible by conventional

fluorescence microscopy above that salt concentration and the solution appeared clear and well-mixed. Next, in order to assess whether TDP-43 forms nanoscale assemblies, we performed smMDS measurements at conditions where no microscopically visible condensates could be detected (i.e., well above $c_{crit}$). To this end, we first performed a continuous scan experiment at 100 mM KCl. The obtained profile, shown in Fig. 8c (top panel), exhibited a broad spread signature, which is characteristic for bulk monomeric protein. Sizing of the profile (Fig. 8c, bottom panel) yielded a hydrodynamic radius of $R_H = 4.29 \pm 0.8$ nm, which is in agreement with the size of monomeric GFP-tagged TDP-43, as predicted by the HullRad model[122]. More importantly, in addition to the characteristic signature for monomeric protein in the continuous scan diffusion profile, we observed bright bursts that were narrowly distributed at the center of the channel on top of the diffusion profile, indicating the presence of clusters (Fig. 8c, top panel). To explore this further, we carried out smMDS step scan measurements. We performed high-resolution step scans with an interval of only 1 μm between steps within the central region of the diffusion profile where clusters would appear. Outside these regions we performed step scans with lower resolution (Fig. 8e, top panel). Clusters were clearly detectable as bursts above the mean bulk signal in the fluorescence time traces (Fig. 8d). These bursts were then counted to give us the number of clusters present at each position in the channel and binned in a histogram (Fig. 8e, bottom panel). Each peak in the histogram was then independently fit to a Gaussian distribution to obtain a mean diffusion distance that could be utilized to calculate $D$ and, thus, $R_H$. Specifically, we extracted the half of the full-width half maximum (FWHM) of the Gaussians as the diffused distance at the four channel positions, corresponding to 0, 10, 26, and 55 s of travel within the channel. The diffusion distances $x$ at each time point $t$ (Fig. 8f) were then fitted with a one-dimensional solution to Fick's second law

$$x \approx \sqrt{2Dt} \qquad (1)$$

to extract $D$ and, thus, $R_H$ via the Stokes–Einstein relation. We performed this analysis with three independent measurements and obtained an average $R_H$ of $120 \pm 10$ nm for TDP-43 nanoclusters (Fig. 8f). We note that the sizes of clusters are similar to FUS clusters previously observed[118], thus indicating that TDP-43 forms similar pre-phase separation clusters as FUS.

The simultaneous, yet independent measurement of the sizes of nanoclusters and monomeric protein further allows estimating the number of monomer units per nanocluster. This is done through comparison of the volume ratios of monomeric TDP-43 and the clustered form. Assuming no restructuring of the protein within the nanocluster, a single cluster could contain as much as 20,000 proteins if the cluster is composed of pure protein. However, as condensates are liquid in nature and contain solvent molecules, typical volume fractions of proteins within condensate systems are on the order of ~10–35%[123–125]; hence, we expect the number of proteins per cluster to be in the range of 2000–7000. In addition to size measurements, the ability to directly count clusters in a digital manner also enables the quantification of cluster particle concentrations and volume fractions. From three repeat measurements, we detected an average number of clusters of $N = 2281 \pm 929$. Using a previously established conversion strategy[60], this corresponds to a flux of $F = 72,606 \pm 29,570$ clusters per second or a cluster particle concentration of $c = 7.24 \pm 2.9$ pM, corresponding to a total nanocluster volume fraction of $\phi = 3.16 \cdot 10^{-5}$. The concentration of TDP-43 nanoclusters detected here was therefore more than an order of magnitude higher than previously determined for FUS nanoclusters formed under the same protein and salt concentrations[118], suggesting a difference in the intermolar interactions that stabilize TDP-43 nanoclusters. Notably, TDP-43 is prone to aggregation and possesses a disordered domain capable of forming

amyloid fibrils[126]. These characteristics may contribute to enhanced intermolecular interactions also in the clustered state, and potentially also explain the higher propensity of TDP-43 to form nanocluster assemblies.

Taken together, we have shown here that smMDS constitutes a solution-based biophysical analysis approach able to size pre-phase separation nanoclusters that are undetectable by conventional fluorescence microscopy. This places smMDS alongside other advanced microscopy techniques like super-resolution microscopy, which are invaluable for visualizing nanoclusters within cellular environments[127–129], and mass photometry for label-free characterization of nanoclusters[130]. Moreover, the discovery of TDP-43 nanoclusters and the understanding of the nature of such sub-diffraction assemblies is critical, in particular, for progressing our understanding of macroscopic phase separation phenomena. Our single-molecule sizing approach therefore provides insight into a largely unexplored area of protein assembly, taking advantage of the capability to elucidate properties of low abundance nanoscale species present in a biomolecular condensate system.

## Discussion

The physical characterization of the sizes, interactions, and assembly states of proteins is vital for a heightened understanding of the biological function, malfunction, and therapeutic intervention of proteins. Of particular interest are insights into the compositional heterogeneity of proteins and protein mixtures such as protein oligomers, protein aggregation, and protein condensate systems. However, such characterization is challenging to achieve using conventional biophysical approaches as these methods are mostly reliant on ensemble readouts, which limits sensitivity and the information content that can be retrieved from such measurements. The smMDS approach developed herein addresses these challenges and takes advantage of the high sensitivity afforded by single-molecule detection and the physical features of diffusion in the microfluidic regime to enable digital sizing of proteins, protein assemblies, and heterogenous multicomponent protein systems directly in solution and in a calibration-free manner.

With different examples, ranging from single proteins to protein assemblies, we have illustrated how the digital nature of the smMDS approach enables diffusional-sizing-based monitoring of protein hydrodynamic radii down to the femtomolar concentration range, thereby pushing the limits of diffusional sizing experiments by more than 5 orders of magnitude. Our study further demonstrates that the size range measurable by smMDS spans from under 1 nm to over 100 nm, aligning with the size spectrum typically covered in standard bulk MDS experiments. smMDS further enables the measurement of binding affinities of protein interactions at the single-molecule level and allows resolving high- and low-oligomeric states of proteins to gain insights into subpopulation distributions and oligomer equilibria. We further characterized the polydisperse nature of protein assemblies on the example of a protein aggregation reaction and applied smMDS to discover nanoclusters of a phase-separating condensate system. These examples highlight the capability of the approach to elucidate properties of low abundance species present in heterogeneous biomolecular systems.

The smMDS platform, as implemented in our study, combines microchip-based diffusional sizing with single-color scanning confocal microscopy. Future iterations of the platform could also incorporate, for example, multicolor single-molecule detection and FRET techniques[131–133], as well as other downstream microfluidic separation modalities[134]. These advancements would further improve sensitivity, resolution, and the depth of information gathered. We also anticipate that the smMDS platform is adaptable for multiplexing across various conditions, setting the stage for high-throughput analysis. In this context, we also foresee the potential evolution of smMDS into a commercial benchtop instrument, broadening its application and accessibility. It is also important to note that the utility of smMDS is not confined to the size range between 1–100 nm. It is particularly adaptable for larger sizes[47,135,136], allowing the quantification of larger proteins and protein assemblies, or their interaction with other larger biomolecular or supramolecular assemblies such lipid vesicles or nanoparticles, through modifications in the chip design and adjustments in the flow rates.

The many examples presented throughout this study clearly showcase the potential of smMDS in characterizing the sizes, interactions, and assembly states of proteins. However, it is also important to acknowledge its inherent limitations. As with other single-molecule fluorescence methods, smMDS relies on protein labeling, a process that not only adds extra steps to the experimental workflow but also introduces the possibility that fluorescence labels could influence protein interactions. Furthermore, effective labeling often requires nano- or micromolar protein concentrations, which can be a significant constraint when working with proteins that are either scarce or unavailable in large quantities. Another challenge is the requirement for specialized expertise in microfluidics and single-molecule optics, as well as substantial investment in equipment, which may be a barrier to broader adoption until a dedicated instrument becomes available. Finally, it is essential to recognize that smMDS, as a powerful tool for in vitro protein analysis, is limited to probing analytes in aqueous buffer solutions. This is in contrast to other single-molecule methods like fluorescence correlation spectroscopy and super-resolution microscopy, which can be applied to live cell environments. Nonetheless, smMDS could be potentially adapted for effective use in a range of complex biological settings. Our previous work with bulk MDS has demonstrated its effectiveness in measuring protein sizes and interactions in environments such as serum or cell lysate[42,46,53]. Building on this foundation, we are optimistic about the adaptability of smMDS for analyzing proteins in diverse biological matrices and clinical samples.

In recent years, a number of sizing techniques have evolved that, similarly to smMDS, operate with minimal sample requirements and sensitivities down to the single-molecule regime. For example, mass photometry[32] (also known as interferometric scattering microscopy)[33–35] has revolutionized our ability to measure protein mass in solution without labeling, offering a unique perspective on protein heterogeneity. However, mass photometry may encounter specificity challenges in complex samples owing to its label-free nature and tends to be more effective for characterizing larger proteins and assemblies, limitations that smMDS is designed to address. In parallel, super-resolution microscopy has provided unprecedented insight into the spatial organization and interactions of proteins at the nanoscale[30,31]. These approaches can be equally applied to in vitro as well as live cell environments. However, super-resolution techniques typically require complex image reconstruction and data analysis processes, which can lead to artifacts and biases, issues less prevalent with smMDS. Also, well-established single-molecule techniques such as fluorescence correlation spectroscopy[22,23], brightness analysis[24,25], and photobleaching step analysis[26,27] have significantly contributed to the advancement of quantitative protein analysis both in vitro and in vivo. However, these methods come with inherent limitations, including the need for calibration to determine absolute sizes. In contrast, smMDS efficiently circumvents this challenge by offering a calibration-free approach.

Taken together, the capabilities of smMDS augment the information content from sizing experiments beyond what is achievable and assayable with classical techniques. smMDS not only enables direct digital sizing of proteins and protein assemblies, but also provides quantitative information on protein interactions and heterogenous multicomponent protein systems. Given the key features of the technique, we anticipate that the smMDS approach will have a

multitude of applications in quantifying the sizes and interactions of proteins and other biomolecules in various areas of biological and biomedical research, including the mechanistic and functional analysis of proteins, the molecular design of protein therapeutics, and the characterization of nanomedicines and biomaterials.

## Methods

### Protein and sample preparation

Alexa 488 carboxylic acid was obtained as lyophilized powder from Thermo Fisher. Stock solutions at millimolar concentrations were prepared in dimethyl sulfoxide and further diluted in phosphate buffered saline (PBS) buffer. Lysozyme, thyroglobulin, HSA, and RNase A were purchased from Sigma Aldrich as lyophilized powder in the highest purity available and suspended in 100 mM $NaHCO_3$ buffer (pH 8.2). Human Leukocyte antigen (HLA) A*03:01 was obtained through the NIH Tetramer Core Facility at Emory University (USA) and rebuffered into 100 mM $NaHCO_3$ buffer (pH 8.2). The HLA-antibody W6/32 (mouse monoclonal anti-HLA Class I antibody [W6/32], isotype: IgG2a, host species: mouse, clonality: monoclonal, clone: W6/32, concentration: 1 mg/mL, Cat#: ab22432) was obtained from Abcam. Human wildtype α-synuclein was recombinantly produced following a protocol detailed elsewhere and prepared in PBS[137]. TDP-43 was produced as a C-terminal EGFP-tagged protein variant in insect cells following previously published procedures[115] and stored in 50 mM Tris-HCl (pH 7.4), 500 mM KCl, 5% (w/v) glycerol, 1 mM DTT. Concentrations were determined by UV/VIS spectroscopy.

### Protein labeling

Protein solutions (lysozyme, thyroglobulin, HSA, RNase A, HLA, and α-synuclein) were mixed with an excess of N-hydroxysuccinimide (NHS)-ester-functionalized Alexa 488 dye (Thermo Fisher). The dye-to-protein ratios were optimized to maintain an average degree of labeling (DOL) below one. Specifically, the ratios were: 1:1 for thyroglobulin, 2:1 for lysozyme, HSA, RNase A, and HLA, and 3:1 for α-synuclein. Details of the protein concentrations used for labeling are given in Supplementary Table 3. The labeling reaction was incubated for one hour at room temperature, except for α-synuclein, which was incubated at 4 °C for 16 h. Subsequently, labeled proteins were separated from unbound dye using size-exclusion chromatography using an AKTA pure chromatography system (Cytiva) with PBS as elution buffer. A Superdex 200 increase 10/300 GL column (Cytiva) was used for all proteins except for α-synuclein, for which a Superdex 75 increase 10/300 GL column (Cytiva) was employed. Post-separation, the concentrations of the labeled proteins were determined via UV/VIS spectroscopy. The concentrations of labeled protein and the labeling efficiencies, in terms of DOL, are reported in Supplementary Table 3. The proteins were stored at −80 °C until further use.

### Generation of labeled α-synuclein oligomers and fibrils

A cysteine-containing variant (N122C) of α-synuclein was used for the preparation of labeled α-synuclein oligomers. The protein variant was produced following previously published procedures[77] and labeled with maleimide-functionalized Alexa 488 dye, followed by purification using Sephadex G25 column (GE Healthcare). Oligomers were produced according to procedures detail elsewhere[77]. Briefly, labeled monomeric α-synuclein was lyophilized in deionized water and subsequently resuspended in PBS (pH 7.4) at a final concentration of ~6 mg mL$^{-1}$. The resulting solution was passed through a 0.22 μm filter (Millipore) before incubation at 37 °C for 16 h under quiescent conditions. Small amounts of fibrillar species were removed by ultracentrifugation. Excess monomeric protein was then removed by multiple filtration steps using 100-kDa cut-off membranes. The final oligomer concentration was determined by UV/VIS spectroscopy.

Recombinant human wildtype α-synuclein was used for the generation of α-synuclein fibrils. Fibrils were prepared from a mixture of Alexa 488 labeled and unlabeled α-synuclein protein, following previously published procedures[47]. Briefly, a mixture containing 10% labeled and 90% unlabeled α-synuclein monomer was aggregated on an orbital shaking incubator (Innova 4400, New Brunswick Scientific) at 37 °C and 200 rpm for 4 days to generate 1$^{st}$ generation fibrils (i.e., seeds). These fibrils were then recovered by centrifugation at $15,000 \times g$. Subsequently, 10% of these fibrils were incubated with 9% labeled and 81% unlabeled monomer on the same orbital shaking incubator at 37 °C and 200 rpm for 3 days to generate 2$^{nd}$ generation fibrils. After centrifugation at $15000 \times g$ and recovery, 2$^{nd}$ generation fibrils were sonicated (10% power, 30% cycles for 90 s) using a Sonopuls HD 2070 ultrasonic homogenizer (Bandelin) and stored at room temperature until further use. Fibril concentration was determined by UV/VIS spectroscopy.

### smMDS platform

The approach described here integrates microchip-based diffusional sizing with confocal fluorescence detection. Schematics of the microfluidic device, the optical setup, and their integration are shown in Fig. 1a and Supplementary Fig. 5. Briefly, the microfluidic chip design is based on previously reported device designs for diffusional sizing[42,47,138]. The device has two inlets, one for the injection of the sample and one for the injection of co-flowing buffer solution. Channels of 25 μm in height and 50 μm in width, respectively, direct the sample and the buffer solutions to an entry nozzle. At the entry nozzle, the sample and the buffer stream merge into an observation channel of 25 μm in height and 225 μm in width, in which diffusion is monitored. Notably, the channel geometry at the nozzle point is designed such that the sample and buffer solution are drawn through the chip in a -1:8 volume ratio. The observation channel is folded multiple times and is approximately 90'000 μm long and terminates at a waste outlet where negative pressure is applied by a syringe pump. Scanning markers are integrated into the chip adjacent to the observation channel for defining the start and end points of the scan trajectory. Details on the fabrication of the device by standard soft-lithography and molding techniques are given below.

The optical unit of the smMDS platform is based on fluorescence confocal microscopy and optimized for microfluidic experiments. The microscope is built around a 'rapid automated modular microscope' (RAMM) frame (Applied Scientific Instrumentation (ASI)) and is equipped with a motorized x,y,z-scanning stage (PZ-2000FT, ASI), onto which the diffusional sizing chip is mounted. For controlling the exact sample placement along the optical axis of the microscope, the stage is equipped with a z-piezo. To excite the sample in the device, the beam of a 488-nm wavelength laser (Cobolt 06-MLD, 200 mW diode laser, Cobolt) is passed through a single-mode optical fiber (P3-488PM-FC-1, Thorlabs) and collimated at the exit of the fiber by an achromatic collimator (60FC-L-4-M100S-26, Schäfter + Kirchhoff) to form a beam with a Gaussian profile. The beam is then directed into the microscope, reflected by a dichroic beamsplitter (Di03-R488/561, Semrock), and subsequently focused to a concentric diffraction-limited spot in the microfluidic channel through a 60x-magnification water-immersion objective (CFI Plan Apochromat WI 60x, NA 1.2, Nikon). The emitted light from the sample is collected via the same objective, passed through the dichroic beam splitter, and focused by achromatic lenses through a 30-μm pinhole (Thorlabs) to remove any out-of-focus light. The emitted photons are filtered through a band-pass filter (FF01-520/35-25, Semrock) and then focused onto a single-photon counting avalanche photodiode (APD, SPCM-14, PerkinElmer Optoelectronics), which is connected to a TimeHarp260 time-correlated single photon counting unit (PicoQuant).

### Fabrication of microfluidic devices

Microfluidic devices for smMDS were fabricated in poly (dimethylsiloxane) (PDMS) using standard soft-photolithography

methods[139]. Briefly, in the first step, the device design was created in AutoCAD (version 2020) software (Autodesk) and printed onto acetate transparencies (Micro Lithography Services), with six layouts arranged within a 3-inch diameter area. Subsequently, a master mold was constructed. This process involved coating a polished silicon wafer (Prime CZ-Si wafer, 3 inches, WSD30381200B1314SNN1, MicroChemicals) with SU-8 3025 photoresist (Kayaku) and spinning it to achieve a thickness of about 25 μm. The printed acetate mask was then positioned on the coated wafer and exposed to UV light using a custom-built LED-based apparatus[140]. After UV exposure, the mold was developed in propylene glycol methyl ether acetate (PGMEA, Sigma Aldrich), creating the final master mold with six device impressions on a single wafer. The exact height was measured by a Dektak profilometer (Bruker). To form PDMS chips, Sylgard 184 silicone elastomer (Dow Corning) was mixed with its curing agent at a ratio of 10:1 (w/w). This mixture was then poured over the master mold placed within a petri dish, degassed to remove air bubbles, and cured for about 1 h at 65 °C. The solidified PDMS was peeled off from the master mold and individual chips were cut out using a scalpel. Access holes for the inlet and outlet connectors were then punched with biopsy punches. The PDMS devices were bonded onto a thin glass coverslip (no. 1.5, Menzel) after both the PDMS and the coverslip glass surfaces had been activated by oxygen plasma (Diener electronic, 40 % power for 30 s). This process forms closed channels with three sides made of PDMS and a fourth of glass. Before injecting buffer and samples into the channels, the microfluidic chips were rendered more hydrophilic through an additional plasma oxidation step (Diener electronic, 80% power for 500 s)[141].

### Experimental procedures

All experiments were performed at room temperature. Buffer was PBS (pH 7.4) in all experiments, except in nanocluster experiments, where TRIS buffer was used (50 mM TRIS-HCl, pH 7.4). Buffers were supplemented with 0.01% Tween 20 (Thermo Fisher) to prevent adhesion of molecules to chip surfaces. The PDMS–glass device was secured to the motorized, programmable microscope stage. Co-flowing buffer and sample were entered into the chip through gel-loading tips inserted into the appropriate inlet orifices and drawn through the chip by applying negative pressure with a syringe (Hamilton) and syringe pump (Cetoni, neMESYS) connected to the outlet. Flow rates were 100 μL/h in all experiments, except in nanocluster experiments, where the flow rates were 150 μL/h for continuous scan experiments and 60 μL/h for step scan experiments. The flow was allowed to equilibrate over six minutes before data acquisition. Diffusion profiles were obtained by translocating the confocal volume either in a continuous or stepwise manner through the four innermost channels of the microfluidic sizing chip using a custom-written Python script that simultaneously controlled the stage movement and the data acquisition at the mid-height of the device (i.e., ~12.5 μm above the surface of the glass coverslip). Continuous scans were performed at 20–100 μm/s. Step scans were done in 200–400 steps for a duration of 1–60 s at each position. The scanning markers were used to define the x, y, z-coordinates of the start and end positions of the scan trajectory. Each experiment was performed in a freshly fabricated PDMS device. The laser power at the back aperture of the objective was adjusted to 370 μW in all experiments, except for experiments on fibrils, where laser powers of 100 μW were used, and for experiments on nanoclusters, where laser powers of 6 μW were used. Photon recordings were done in T2 mode and the arrival times of photons were measured with respect to the overall measurement start with 16-picosecond resolution.

### Data analysis

Data analysis and plotting was done in Python (version 3.6). In continuous scan experiments, photon recordings were binned in 1-ms intervals to obtain intensity readouts, from which diffusion profiles were generated by plotting the obtained fluorescence intensities as a function of chip position. In step scan experiments, diffusion profiles were created by extracting single-molecule events from the recorded time trace at each position using a burst-search algorithm, and plotting the obtained the number of counted molecules as a function of chip position. The custom code (written in Python, version 3.6) is available as Supplementary Software or on the GitHub repository: https://github.com/gkrainer/smMDS. The burst-search algorithm identifies single molecules from the photon time trace by applying a combined $IPT_{max}$ and $N_{min}$ threshold. $IPT_{max}$ and $N_{min}$ were in the range of 0.005–0.02 ms and 5–20 number of photons, respectively, for all experiments performed under single-molecule conditions, unless otherwise stated. In addition, a Lee filter of 2–4 was applied that smoothens regions of constant signal while keeping those with rapid parameter changes unaffected (such as the edges of the bursts).

To extract size information, the obtained diffusion profiles, from both continuous and step scan experiments, were analyzed with a custom-written analysis software. The custom code (written in Python, version 3.6) is available as Supplementary Software or on the GitHub repository: https://github.com/gkrainer/smMDS. This script fits the obtained diffusion profiles with simulated diffusion profiles from numerical model simulations solving the diffusion–advection equations for mass transport under flow (see Supplementary Note 1 for details). A least-squares error algorithm is used to find simulated profiles with the lowest residuals to determine $D$ and recover $R_H$ via the Stokes–Einstein relationship.

### Reporting summary

Further information on research design is available in the Nature Portfolio Reporting Summary linked to this article.

## Data availability

All the data generated in this study are available within the main text and the Supplementary Information file. Source data are provided in this paper.

## Code availability

Computer code used in this article for single-molecule analysis and the analysis of diffusional sizing profiles is available as Supplementary Software or on the GitHub repository: https://github.com/gkrainer/smMDS.

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

## Acknowledgements

The research leading to these results has received funding from the European Union's Horizon 2020 research and innovation programme through the Marie Skłodowska-Curie grant MicroSPARK (agreement no. 841466; G.K.), from the European Union's Horizon 2020 research and innovation programme through the Marie Skłodowska-Curie grant MicroProtLip (agreement no. 896068; M.A.C.), from the European Research Council (ERC) under the European Union's Horizon 2020 research and innovation programme through the grant DiProPhys (agreement no. 101001615; T.P.J.K.), the Herchel Smith Fund of the

University of Cambridge (G.K.), the Wolfson College Junior Research Fellowship (G.K.), the University of Graz (G.K.), the European Innovation Council (EIC) under the European Union's Horizon 2020 Future and Emerging Technologies Open (FET-OPEN) programme through the grant NANOPHLOW (agreement no. 766972; T.P.J.K.), from the European Union's Horizon 2020 research and innovation programme through the grant NANOTRANS (agreement no. 674979; Q.A.E.P., T.P.J.K.), and the Newman Foundation (T.P.J.K.). T.J.W. acknowledges funding from the Harding Distinguished Postgraduate Scholar Programme, administered by the Cambridge Commonwealth, European & International Trust at the University of Cambridge. K.L.S. acknowledges funding from the Schmidt Science Fellowship in partnership with the Rhodes Trust and the St. John's College Junior Research Fellowship. V.K. acknowledges funding from an NIHR Fellowship (PDF-2016-09-065) and as a Paul I. Terasaki Scholar. This work was also supported by a grant from the Engineering and Physical Sciences Research Council (EPSRC) (grant no. EP/L015889/1; R.P.B.J.). The authors would also like to thank the EPSRC Centre for Interdisciplinary Ph.D. Training and Research in Nanoscience and Nanotechnology (NanoDTC).

## Author contributions

G.K. and T.P.J.K. conceptualized the study, G.K. devised the methodology, G.K., M.M.S., T.J.W., J.F., E.A.A., G.S., M.A.C., H.A., L.C. performed the investigation, G.K., R.P.B.J., M.M.S., T.J.W., Q.A.E.P. performed the analysis, R.P.B.J. and Q.A.E.P. contributed relevant software, W.E.A., K.L.S., T.W.H., T.M.F., V.K., S.A., F.-U. H., S.F.L. provided materials and expertise. G.K. wrote the original draft and all other authors reviewed and edited it.

## Competing interests

G.K., K.L.S., W.E.A., and T.P.J.K. declare the following competing interests. Parts of this work have been the subject of a patent application filed by Cambridge Enterprise Limited, a fully-owned subsidiary of the University of Cambridge. Inventors: Krainer, G.; Saar, K.L.; Arter, W.E., Knowles, T.P.J.; Applicant: Cambridge Enterprise Ltd.; Title: Highly sensitive biomolecule detection and quantification. Publication Number: WO/2021/176065; Publication Date: 10.09.2021; International Application No.: PCT/EP2021/055614; International Filing Date: 05.03.2021. The remaining authors declare no competing interests.
