## [Peer Review File · Nature Communications]

Reviewers' Comments:

Reviewer #1:

Remarks to the Author:

This manuscript introduces an innovative technique for microfluidic diffusional sizing (MDS) at the single-molecule level, termed smMDS. The distinct advantage of smMDS over conventional ensemble-based MDS is its capability to distinguish molecular and supramolecular heterogeneities, as typically seen in samples with protein mixtures, oligomers, aggregates, and condensates. The introduction of smMDS marks a significant milestone with potential widespread applications in the realm of biomolecular sciences. Through their work, the authors demonstrate the robustness of smMDS by assessing binding affinities and detecting nanoclusters with high precision. The clarity of the manuscript, complemented by its exemplary illustrations, ensures that the core messages are easily grasped by a broad readership.

To further enhance this already outstanding paper, we suggest the authors address the following points:

(1) Fluorescent labeling using NHS esters produces a varied number of dye molecules per protein, resulting in broad intensity distributions across the protein population. How have the authors accounted for this variation?

(2) It would be beneficial to state the labeling efficiency for each protein discussed in the paper and the specific protein concentrations used for labeling.

(3) The chosen labeling methods (utilizing NHS and maleimide moieties) demand micromolar protein concentrations. This presents a limitation for the smMDS process when dealing with proteins that are scarce or not available in high concentrations. Highlighting this constraint more conspicuously would offer greater clarity.

(4) It would be beneficial if the authors could either articulate the limitations of the proposed method as lucidly as they do for existing techniques or temper the criticisms of existing methods, especially the critique of FCS for overparameterization found on page 4.

(5) Regarding the binding assay executed with labeled HLA and unlabeled W6/32 antibody: would it be feasible to conduct this assay with an unlabeled HLA and labeled antibody? This modification might offer more versatility in affinity profiling, permitting the probing of target molecules within cell extracts without needing to label the analyte.

(6) For a more comprehensive understanding, a brief description of the advection–diffusion model, even if previously published, should be included. This would offer the reader a clearer understanding of the fitting processes and associated parameters.

(7) Expectedly, diffusion profiles in the microfluidic setup should be symmetrical. Yet, deviations in Gaussian fits within the intensity profiles appear systematically skewed. Could the authors provide insight into this phenomenon?

(8) Notably, smMDS seemingly expands the size range MDS can assess, given the measurements of free Alexa 488 dye. This advantageous aspect deserves further elucidation.

(9) The comparative reference measurements of TDP-43 aggregation in wide-field mode should be incorporated into the manuscript.

(10) In Figure 4c, what is the meaning of the positioning of experimental data points along the x-axis, given that this is a bar chart? Could the authors clarify the significance of these positions?

(11) For all mean values presented with standard deviations, it would enhance clarity if the authors maintained a reasonable number of decimal places. For instance, in the caption to Figure 4, it might be clearer to represent the dissociation constant as $K_d = (400 \pm 40) \text{ pM}$.

(12) On page 14, the use of the term "asymmetric Gaussian" is ambiguous since Gaussian functions are inherently symmetric. Could the authors clarify their meaning?

(13) The statement on page 15 regarding the correlation between measured monomer, dimer, and trimer abundances with the equilibrium distribution of serum albumins would benefit from a more quantitative comparison or reference.

(14) Page 22 presents Equation (1) as a one-dimensional approximation to Fick's law. It might be more accurate to describe it as a solution to Fick's second law or the diffusion equation for specific 1D scenarios.

(15) The custom-made microfluidic chip, essential to the study, is not depicted in the manuscript. Displaying the flow path with the chip's dimensions and linking it to the diffusion model will aid comprehension. It will also clarify the flow paths illustrated in Figure 1's panels b and c (or are these meant to be time series?).

(16) A concise explanation of the molding process used and the point at which the resin was cast could offer added clarity.

(17) For the GitHub documentation, revising and merging the Instructions.txt and Readme.md files to incorporate a step-by-step guide with a sample dataset would be invaluable. The current informal tone might not be best suited for publication.

Minor comments:

- Page 5, Figure 1: Please clarify the discrepancy in the dimensions of fit errors between different experiments.

- Page 9, paragraph 3, line 2: Should "confocal VOLUME" be the appropriate term?

- Page 15, Figure 6a,c: Could the authors explain the rationale behind chosen thresholds? Also, does the plot in Figure 7c align with theoretical assumptions regarding alpha-synuclein oligomer aggregation number and size?

- Page 17, Figure 7c: The trace for the monomer is somewhat obscured. Displaying the traces with an offset might enhance clarity.

- Page 17, Figure 7e: Adjusting the scale to better visualize the signal peaks for the monomer would be helpful.

- Page 18: If an excess of one molecular species is present, is there an estimated threshold for the ratio? How might one discern a fraction from the noise, especially when the ratio isn't pre-known?

- Page 22: When referencing the sizing of the profile in Figure 8c's bottom panel, it would be more informative to provide a reference for the assertion made about the hydrodynamic radius.

- alpha-synuclein interacts with anionic phospholipids, which in turn can lead to self-association of the peptide. Did the authors observe any influence of the Alexa 488 dye, which is itself anionic?

Reviewer #2:

Remarks to the Author:

The manuscript describes an improved method that can measure protein size at very low concentrations. This method builds on previous work developing a microfluidics system that measures molecular diffusivity. The ability to measure protein size based on the diffusivity of single proteins is a very powerful technique, and in this case, it is used to demonstrate several potential applications. While the development of this technology is exciting, the manuscript would benefit from a more detailed description of how this method compares to others, rather than just being an improved version of the previously described methods.

The introduction contains little reference to similar work in the literature. Young et al (2018) used a light scattering approach that also extracted biophysical parameters from single molecules, and there are several other instances of single-molecule studies of single proteins.

While the current method has the advantage of requiring little protein, the protein does require labelling. What are the typical labelling efficiencies that are observed for different proteins? Is there any evidence that the presence of the fluorescent tag affects protein interactions? In the case of distinguishing between α -synuclein fibrils and monomers, how is the ability to detect the monomeric species in a background of fibrils enhanced but the high degree of labelling in the monomers compared with the fibrils (only 10% labelled)?

Fig 2c shows that 20 pM HSA appears to give an Rh of 3.29nm when measured by step scan, but 3.63 when measured by continuous scan, however, these data points appear to be reversed in Fig 2d.

Fig 3 shows the range of experimentally derived values for different proteins at 10 pM. In Fig 3a, the experimental RH shown for HSA is $\sim 4.0 \pm 1$ nm. However, the equivalent data shown in Fig 2d gives an experimental value of $\sim 3.75 \pm 0.25$ nm.

Given that the abstract describes "ultrasensitive sizing of proteins down to the femtomolar concentration" what is the accuracy and reproducibility of the technique for measuring protein size? The values of $\sim 4.0 \pm 1$ nm for HSA (Fig 3a) equate to a MW of 102 kDa and a range of 42-202 kDa when using the online converter at <https://www.fluidic.com/toolkit/hydrodynamic-radius-converter/> which is higher than the expected MW of 66kDa. Similarly, the value of $\sim 2.5 \pm 1$ nm for RNase (expected MW = 14 kDa) equates to a MW of 24 kDa, with a range of 5-67 kDa.

HSA is also used as a model protein to show that smMDS can be used to resolve protein oligomeric states (Fig 5). In this case, the monomeric protein has an RH of 3.46nm at 10pM, compared to a value of 3.77nm at 10pM previously described in Fig 2c. Is this difference due to a global analysis of the data in Fig 2c, compared to the deconvolution analysis in Fig 5? How does the proportion of monomer/dimer/trimer/tetramer compare to that determined by other methods? It is stated that "These values are in very good agreement with the expected equilibrium distribution of serum albumins(ref 70)", but the data presented in ref 70 for BSA does not have a detailed distribution of species. Are the oligomeric species of HSA in equilibrium? The methods state that the labelled protein was passed down a SEC column to separate the unbound dye, which also should have separated the oligomeric species. If the distribution is under equilibrium, it may be expected to observe a higher proportion of smaller species at lower concentrations and an increased population of trimers and tetramers at higher protein concentrations. For this (and other measurements) it may be useful to compare how smMDS compares to other methods. Techniques such as analytical ultracentrifugation can be used at pM concentrations using fluorescently labelled protein.

In terms of investigating the interaction between HLA and W6/32, the authors compare the result of using step mode scanning compared to their previous work using continuous scanning (Schnieder et al 2023). Comparison of the data suggests that the use of step mode has allowed the inclusion of an additional data point at 10 pM, but the data for the two different methods

appear to be similar – perhaps more discussion could be included as to the advantages afforded by the step mode method.

It would be useful to include more discussion regarding how the current results compare to other methods. For example, the Rh range observed for α -synuclein oligomer subpopulations is compared to a different study that used AFM (ref 70). What are the comparisons of the different analyses? AFM also detects single particles.

Reviewer #3:

Remarks to the Author:

Reviewer #4:

Remarks to the Author:

In this manuscript, Krainer et al. developed a method single-molecule microfluidic diffusional sizing (smMDS), a variant of MDS that exploits “single-molecule” fluorescence detections in a tight excitation volume to characterize purified protein/protein complexes/simple protein mixtures at pico-to-femtomolar concentrations. The authors demonstrated the application of smMDS in quantifying the hydrodynamic radii (Rh), pairwise interaction affinities (Kd), and assembly and oligomeric states of specific proteins. smMDS features innovative experimental designs and enables protein sizing characterizations at protein concentrations 5 orders of magnitude lower than what the conventional MDS is capable of characterizing. However, there are other established single-molecule techniques capable of quantifying the dimensions, interaction affinities, and oligomeric states of proteins in vitro. A prominent example (for interaction affinity and oligomeric state measurements) is single molecule mass photometry, which is based on precise measurements of light scattering of protein molecules. Another example (for dimension measurements) is super-resolution fluorescence microscopy. A comparison between these existing methods and smMDS in measuring the same protein/protein complex properties is missing in the current manuscript. Without this comparison, it is difficult for readers to evaluate the significance and unique strength of smMDS and when to apply smMDS instead of other techniques for protein characterizations. This and other comments are specified below.

1. As summarized above, single molecule mass photometry has been shown to successfully characterize the oligomeric states of proteins in vitro as well as quantify the dissociation constant (Kd) of protein complexes (PMID: 29700264; PMID: 33068635; PMID: 34765909). It is important to discuss how smMDS compares with single molecule mass photometry in measuring these protein properties.

2. It is unclear how sensitive smMDS is in distinguishing different oligomeric states of a protein. In Figure 5a, the authors fit the fluorescence intensity distribution to a four-component model (one “asymmetrical Gaussian” distribution and three Gaussian distributions) because they have a priori knowledge about the oligomeric states of the protein HSA. The overall shape of the fluorescence intensity histogram does not show four distinct peaks. It would be helpful if the authors can show with statistical power that their model is capable predicting the four oligomeric states for HSA (rather than some other number) via Bayesian/Akaike information criterion or some other model comparison method. In other words, can they demonstrate that if they did not know that HSA had four oligomeric states, they would be able to extract that information and the correct states? It would also strengthen the authors’ conclusions if they can use smMDS to reveal the oligomeric states and their fraction in a solution for a protein whose oligomerization has not been previously

characterized and then verify their smMDS findings on oligomeric states using orthogonal approaches.

A related question: Which "asymmetrical Gaussian" did the authors use? There are many asymmetrical Gaussian distributions (skew normal etc.) and it would be helpful to see one line in the supplemental info listing the actual CDF/PDF they fit to.

3. In their characterization of the nanoscale clusters of GFP-tagged TDP-43, the authors concluded that the clusters have narrowly distributed sizes, with an Rh of 121.5+/-14.5 nm. This is an interesting finding as a phase separation mixture usually have droplet sizes more widely distributed. In this pre-phase separating regime, can the authors provide alternative evidence that verifies the uniformly distributed cluster sizes? The authors claimed that under the condition of their smMDS measurements, the clusters were undetectable by conventional microscopy. This is puzzling. Although sub-diffraction limit features "cannot be resolved" by conventional microscopy as the authors correctly pointed out, this should only mean that precise quantification of the cluster dimensions is infeasible. The clusters themselves should nevertheless be visible and detectable by microscopy. A cluster with a diameter of ~223 nm contains 2000 – 7000 labelled protein molecules as estimated by the authors and should be easily visible by fluorescence microscopy in principle. It is suggested that the authors clarify this point.

4. Related to the above point, although conventional microscopy cannot resolve sub-diffraction limit features of nanoscale protein clusters, many well-established super-resolution microscopy techniques, e.g., SIM, PALM, and STORM, are capable of characterizing nanoscale features in numerous systems in vitro and in live cells. How does smMDS compare with super-resolution microscopy in characterizing the sizes of pre-phase separating protein clusters?

5. There is not a clear distinction in the background made between the advantages and shortcomings of in vitro vs live-cell measurements. As such, it leads to some relatively one-sided statements about smMDS, which could be somewhat misleading to readers. For example, the authors stated that smMDS is superior to other techniques, e.g., FCS, because the latter relies on calibration for size determination and suffers from overparameterization. Although FCS requires calibration as the authors have correctly pointed out, this current work did not show that smMDS is not prone to overparameterization (convincing statistics showing this are notably absent, as detailed in Point 2). Besides, the authors did not address the critical point that FCS and variants can be used to make size/population measurements in live cells, which is an advantage over smMDS. While buffers like PBS somewhat approximate the cellular conditions, they are far from physiological cellular environment with the presence of numerous other small molecules and macromolecules that could possibly affect the conformation, interaction behaviors, and oligomerization of a protein of interest. It is suggested that the authors directly address not only the advantages of smMDS but also its caveats to improve the quality of the paper.

RESPONSE TO REVIEWERS' COMMENTS (NCOMMS-23-31470)

Reviewer #1:

This manuscript introduces an innovative technique for microfluidic diffusional sizing (MDS) at the single-molecule level, termed smMDS. The distinct advantage of smMDS over conventional ensemble-based MDS is its capability to distinguish molecular and supramolecular heterogeneities, as typically seen in samples with protein mixtures, oligomers, aggregates, and condensates. The introduction of smMDS marks a significant milestone with potential widespread applications in the realm of biomolecular sciences. Through their work, the authors demonstrate the robustness of smMDS by assessing binding affinities and detecting nanoclusters with high precision. The clarity of the manuscript, complemented by its exemplary illustrations, ensures that the core messages are easily grasped by a broad readership.

To further enhance this already outstanding paper, we suggest the authors address the following points:

Reply: We are delighted about the enthusiastic feedback and the remarks on further improving the manuscript. Below we have prepared a point-by-point response to address the reviewer's comments and valuable suggestions.

(1) Fluorescent labeling using NHS esters produces a varied number of dye molecules per protein, resulting in broad intensity distributions across the protein population. How have the authors accounted for this variation?

Reply: Thank you for raising this critical point regarding our use of NHS esters in fluorescent labeling. We are aware that this method can potentially lead to a varied number of dye molecules per protein. To address this, we have optimized the labeling protocol to ensure that the average degree of labeling (DOL) remains below one, primarily by tuning the dye-to-protein ratio. Indeed, our quantification for each protein used in our study verified that the DOL remained consistently below one, as shown in the provided table below. This assures that, on a per-molecule basis, our proteins are predominantly either unlabeled or singly labeled, thereby maintaining a uniform fluorescence intensity across all detected protein molecules. To provide clarity on this aspect, we have included the mentioned table in the Supplementary Information (now Supplementary Table 3, page 9). Furthermore, we have added a more detailed explanation of our labeling procedure and the steps taken to control the DOL in the Methods section of our manuscript (please refer to page 29).

Supplementary Table 3

Protein	Concentration used for labeling	Molar excess dye	Degree of labeling (DOL)	Concentration after labeling and purification
RNase A	20 μ M	3x	0.90	11.8 μ M
HSA	20 μ M	3x	0.99	11.9 μ M
Lysozyme	40 μ M	3x	0.34	24.5 μ M
HLA	3.12 μ M	6x	0.24	734 nM
Thyroglobulin	6.45 μ M	5x	0.34	2.48 μ M
α -Synuclein	120 μ M	3x	0.67	45.6 μ M

(2) It would be beneficial to state the labeling efficiency for each protein discussed in the paper and the specific protein concentrations used for labeling.

Reply: Thank you for this suggestion. As elaborated in our previous response, we have quantified the labeling efficiency, expressed as the degree of labeling (DOL), along with the

specific protein concentrations used for labeling, the molar excess of dye used, and the protein concentration after labeling and chromatographic purification. This data is detailed in the table referenced above. We have updated the manuscript to incorporate this table into the Supplementary Information (now Supplementary Table 3, page 9).

(3) The chosen labeling methods (utilizing NHS and maleimide moieties) demand micromolar protein concentrations. This presents a limitation for the smMDS process when dealing with proteins that are scarce or not available in high concentrations. Highlighting this constraint more conspicuously would offer greater clarity.

Reply: Thank you for pointing out this important aspect. Indeed, our chosen labeling methods do require micromolar protein concentrations. This can be a limitation, especially when working with proteins that are scarce or not readily available in high concentrations. Acknowledging the significance of this constraint, we have taken your suggestion into account and have highlighted this limitation in the Discussion section of the revised manuscript (page 27).

(4) It would be beneficial if the authors could either articulate the limitations of the proposed method as lucidly as they do for existing techniques or temper the criticisms of existing methods, especially the critique of FCS for overparameterization found on page 4.

Reply: Thank you for your constructive feedback. We acknowledge the importance of presenting a balanced view of both the strengths and limitations of our method. Addressing your concern, we have added a section to our manuscript to outline these limitations in the Discussion section more clearly (please refer to page 27–28). Additionally, we have moderated our critique of existing methods like FCS in the Introduction section, specifically the point about overparameterization, and added references (please refer to page 4).

(5) Regarding the binding assay executed with labeled HLA and unlabeled W6/32 antibody: would it be feasible to conduct this assay with an unlabeled HLA and labeled antibody? This modification might offer more versatility in affinity profiling, permitting the probing of target molecules within cell extracts without needing to label the analyte.

Reply: Conducting the assay with unlabeled HLA and labeled W6/32 antibody is indeed technically feasible, especially if probing the target antigens is the primary focus. The reason we opted for labeled HLA and unlabeled W6/32 antibody in our study was primarily driven by the clinical application of detecting HLA antibodies in patients, particularly in the context of organ transplantation. This approach aligns with the methodologies discussed in our referenced HLA paper (Schneider et al. 2023, ref. 54), which focuses on such clinical scenarios. We have added a note about this to the paper for clarity (please refer to page 14). Your suggestion, however, to reverse the binding assay is insightful and presents a direction that could be explored in future studies for broader applications outside of this specific context, a possibility we have now briefly addressed in our manuscript (refer to page 14).

(6) For a more comprehensive understanding, a brief description of the advection–diffusion model, even if previously published, should be included. This would offer the reader a clearer understanding of the fitting processes and associated parameters.

Reply: Thank you for your valuable suggestion. To facilitate a more comprehensive understanding, we have now incorporated a brief but detailed description of the advection–diffusion model in the Supplementary Information (refer to page 2).

(7) *Expectedly, diffusion profiles in the microfluidic setup should be symmetrical. Yet, deviations in Gaussian fits within the intensity profiles appear systematically skewed. Could the authors provide insight into this phenomenon?*

Reply: Thank you for highlighting this aspect. We recognize that occasionally, minor offsets are observed in the diffusion profiles from the channel centers. These offsets are likely due to slight imperfections in the upstream flow path, causing a shift in the peak positioning to either the left or right. Given that our analysis protocol is designed to generate fits within the channel, such upstream variations can result in visible offsets. However, it is important to note that the global fitting procedure we utilize for extracting diffusion constants is resilient towards anomalous shifts that occur within channels, ensuring that the overall analysis remains accurate. We have included a few sentences in our manuscript to clarify this point (please refer to page 10).

(8) *Notably, smMDS seemingly expands the size range MDS can assess, given the measurements of free Alexa 488 dye. This advantageous aspect deserves further elucidation.*

Reply: Thank you for raising this point. It is important to note that the determining factor for the size range that MDS can assess lies in the chip design and/or the used flow rates rather than whether the analysis is conducted at the bulk or single-molecule level. Both MDS and smMDS, in their most common implementation, are equally capable of analyzing biomolecular analytes with R_h ranging from less than 1 nm to over 100 nm. This includes the ability to measure small organic fluorophores like Alexa 488 dye, which fall into the sub-nanometer size category. In our current study, we have successfully measured these with smMDS. Similarly, such measurements have been previously achieved using bulk MDS, as detailed in research papers like Scheidt et al. 2020 (<https://doi.org/10.1039/D0LC00219D>) and Paganini et al. 2021 (<https://doi.org/10.1002/adhm.202100021>). We have incorporated mentions of these insights and further discussion on this topic in our manuscript (please refer to pages 6 and 27).

(9) *The comparative reference measurements of TDP-43 aggregation in wide-field mode should be incorporated into the manuscript.*

Reply: Thank you for your valuable suggestion. In Figure 8b, we studied the phase behavior of GFP-tagged TDP-43 as a function of KCl concentration using wide-field microscopy imaging. Microscopically visible condensates (1–2 μm in diameter) appear below a critical KCl concentration (c_{crit}) of ~50 mM, and condensates decrease in size as the critical concentration is approached. No microscopically visible condensates are detectable above c_{crit} and the solution appears clear and well-mixed. The phase diagram on the left was generated from measurements at five KCl concentrations and a protein concentration of 0.5 μM . Representative widefield fluorescence microscopy images at 100 mM and 25 mM KCl are shown on the right. These images clearly illustrate the concentrations at which TDP-43 phase separation is detectable using wide-field microscopy, as well as the concentrations where phase separation is not observable. We have now revised Figure 8b (please refer to page 23) to clearly indicate that it includes wide-field microscopy images, and we have labeled each of the two representative images with its corresponding KCl concentration to enhance clarity and facilitate easier understanding.

(10) *In Figure 4c, what is the meaning of the positioning of experimental data points along the x-axis, given that this is a bar chart? Could the authors clarify the significance of these positions?*

Reply: Thank you for bringing this to our attention. In Figure 4c, the positioning of the experimental data points along the x-axis is determined by a plotting routine in Python, designed to scatter them randomly. This approach is taken to avoid overlap, thereby improving

the clarity and readability of the chart. This distribution method is chosen purely for visual effectiveness and does not reflect any particular significance pertaining to the data. To ensure clarity for our readers, we have added a sentence to the figure caption of Figure 4, explicitly explaining this aspect of our data presentation. Please refer to page 13 for this addition.

(11) For all mean values presented with standard deviations, it would enhance clarity if the authors maintained a reasonable number of decimal places. For instance, in the caption to Figure 4, it might be clearer to represent the dissociation constant as $K_d = (400 \pm 40)$ pM.

Reply: Thank you for your feedback. In response, we have adjusted the decimal presentation (i.e., significant digits) of mean values and standard deviations across the manuscript for consistency, including the updated notation of the dissociation constant in Figure 4d to $K_d = 400 \pm 40$ pM and in Figure 8f to $R_H = 120 \pm 20$ nm. For details on these changes, please refer to the respective figures on page 13 and page 23.

(12) On page 14, the use of the term "asymmetric Gaussian" is ambiguous since Gaussian functions are inherently symmetric. Could the authors clarify their meaning?

Reply: Thank you for raising this point. You are correct in noting that Gaussian functions are inherently symmetric. In our manuscript, when referring to "asymmetric Gaussian", we actually employed the skew normal distribution, which allows for asymmetry. We have clarified this terminology in the revised version of our manuscript to avoid any confusion (please refer to page 16).

(13) The statement on page 15 regarding the correlation between measured monomer, dimer, and trimer abundances with the equilibrium distribution of serum albumins would benefit from a more quantitative comparison or reference.

Reply: Thank you for the suggestion. Our study found the relative abundances for human serum albumin species to be 67.9% monomer, 20.7% dimer, 7.2% trimer, and 4.2% tetramer (see Figure 5c). These figures are broadly consistent with other serum albumins, like those reported by Young et al. 2018 for bovine serum albumin (88.63% monomer, 9.94% dimer, 1.18% trimer, 0.25% tetramer). We have included these comparisons in our revised manuscript for a clearer quantitative analysis (please refer to page 17).

(14) Page 22 presents Equation (1) as a one-dimensional approximation to Fick's law. It might be more accurate to describe it as a solution to Fick's second law or the diffusion equation for specific 1D scenarios.

Reply: Thank you for your insightful suggestion. In light of your feedback, we have revised the description to accurately represent Equation (1) as a one-dimensional solution to Fick's second law (please refer page 25).

(15) The custom-made microfluidic chip, essential to the study, is not depicted in the manuscript. Displaying the flow path with the chip's dimensions and linking it to the diffusion model will aid comprehension. It will also clarify the flow paths illustrated in Figure 1's panels b and c (or are these meant to be time series?).

Reply: Thank you for this suggestion. We acknowledge the importance of visualizing the custom-made microfluidic chip, which is a crucial component of our study. To enhance understanding and provide clarity, we have included a detailed scheme of the chip in the Supplementary Information (now Supplementary Figure 5, page 8), showing the flow path along with the chip's dimensions. This also serves to clarify the flow paths illustrated in Figure 1's panels b and c.

(16) A concise explanation of the molding process used and the point at which the resin was cast could offer added clarity.

Reply: Thank you for your feedback, and we apologize if our initial explanation of the molding process was unclear. While we did include details in our original manuscript, it seems they were not as clear as intended. To address this, we have revised the relevant section to offer a more detailed and concise explanation of the molding process, including the point at which the resin was cast. Please refer to the updated description on page 31 in the Methods section for this enhanced clarity.

(17) For the GitHub documentation, revising and merging the Instructions.txt and Readme.md files to incorporate a step-by-step guide with a sample dataset would be invaluable. The current informal tone might not be best suited for publication.

Reply: Thank you for your insightful feedback. In response, we've established a dedicated GitHub page for smMDS, accessible here: <https://github.com/gkramer/smMDS/>. We've reorganized the folder layout and refreshed the documentation based on your suggestions. The updated documentation features an extensive step-by-step guide, enriched with an example dataset for enhanced clarity. Additionally, we've refined the documentation's tone to better align with publication standards.

Minor comments:

- Page 5, Figure 1: Please clarify the discrepancy in the dimensions of fit errors between different experiments.

Reply: Thank you for highlighting this aspect. We report two types of errors. The first, represented by the green plots in the diffusion profiles, depicts the local radius error at each profile position and serves to provide a measure of how well the global fit describes the experimental profiles. This error is calculated as the difference between the hydrodynamic radius derived from the global fit and that obtained from the best matching profile at that specific position. Notably, in continuous scanning mode or under higher concentration conditions, increased distortions, especially at the profile's base, lead to a greater observed error. This accounts for the discrepancies in error values observed between bulk and single molecule sizing experiments. The second type of error we report is the error range for the hydrodynamic radius derived from the global fit. This error, determined through a Taylor expansion of the least-square fit and through error propagation, is provided alongside the extracted hydrodynamic radius, indicated within square brackets. We have expanded the legend of Figure 1 (please refer to page 6) to include additional information, offering a more detailed explanation of the error depicted in the figure. For additional information, please refer to the section titled "Analysis of Diffusion Profiles Using the Advection–Diffusion Model" in the Supplementary Information.

- Page 9, paragraph 3, line 2: Should "confocal VOLUME" be the appropriate term?

Reply: Thank you for your attention to detail. We have revised the text to correctly use the term "confocal volume" as suggested (please refer to page 10).

- Page 15, Figure 6a,c: Could the authors explain the rationale behind chosen thresholds? Also, does the plot in Figure 7c align with theoretical assumptions regarding alpha-synuclein oligomer aggregation number and size?

Reply: Thank you for your inquiries regarding Figure 6a and c. In our analysis, we explored a range of photon thresholds, specifically between 5 and 50 total photons, to ensure that the chosen thresholds were representative of the intensity variations observed in oligomer burst

signals. We have included this rationale for choosing thresholds in the manuscript for clarity (please refer page 18). Regarding Figure 7c (we assume the reviewer did actually refer to Figure 6c, as this is about α -synuclein oligomers), the R_H distribution observed in our smMDS study aligns well with results from extensive research on α -synuclein oligomers using various methods. For a detailed comparison, please see our response to the final query of Reviewer #2. We have now broadened our discussion in the revised manuscript to encompass these studies (please refer to page 18).

- Page 17, Figure 7c: *The trace for the monomer is somewhat obscured. Displaying the traces with an offset might enhance clarity.*

Reply: Thank you for your suggestion regarding Figure 7c. The trace depicted is a representative fluorescence time trace from a step scan measurement. This measurement includes a mixture of α -synuclein fibrils (9 nM monomer equivalent) and monomeric α -synuclein (1 nM), with an intensity threshold applied to distinguish signals from fibrils (red) and the monomer (purple). As both the signals from fibrils and the monomer originate from the same raw data, applying an offset to the monomer signal might lead to misinterpretation of the data. Consequently, we have opted not to adjust the presentation in this manner.

- Page 17, Figure 7e: *Adjusting the scale to better visualize the signal peaks for the monomer would be helpful.*

Reply: Thank you for this suggestion. The trace depicted is the monomer control. We kept the scale the same as in Figure 7f to make the difference more clearly visible. Consequently, we have opted not to adjust the presentation in this manner.

- Page 18: *If an excess of one molecular species is present, is there an estimated threshold for the ratio? How might one discern a fraction from the noise, especially when the ratio isn't pre-known?*

Reply: Thank you for your query. In our analysis in Figure 7c, we distinguished the bulk monomer signal from the fibril burst signals using an intensity threshold approach. Specifically, fibrils were identified as bursts with a fluorescence count rate exceeding 250 kHz, following application of a Savitzky-Golay smoothing filter. The residual signal, representing the mean bulk signal of fluorescence time traces in the absence of fibril signals, is attributed to the monomer. For our analysis, selecting such an arbitrary threshold proved effective. Alternatively, one could fit the Savitzky-Golay filtered monomer signal with a Poisson or Gaussian distribution to define its baseline and spread. A threshold based on this distribution's standard deviation, such as 3 or 5 sigma, could then effectively differentiate monomer from fibril species.

- Page 22: *When referencing the sizing of the profile in Figure 8c's bottom panel, it would be more informative to provide a reference for the assertion made about the hydrodynamic radius.*

Reply: Thank you for your insightful observation. The hydrodynamic radius of GFP-tagged TDP-43 mentioned in the context of Figure 8c's bottom panel is a predicted value, calculated using the HullRad model by Fleming & Fleming 2018 (<https://doi.org/10.1016/j.bpj.2018.01.002>). To provide greater clarity and substantiate this assertion in the manuscript, we have now included an appropriate reference to this model (please see page 25).

- *alpha-synuclein interacts with anionic phospholipids, which in turn can lead to self-association of the peptide. Did the authors observe any influence of the Alexa 488 dye, which is itself anionic?*

Reply: Thank you. We did not observe any effect of the anionic nature of the Alexa 488 dye on α -synuclein self-association. The hydrodynamic radius we measured for the monomeric, Alexa 488-labeled α -synuclein is consistent with the expected dimensions of a fully unfolded 15 kDa protein. This observation suggests that there is no significant oligomerization attributable to the presence of the Alexa dye.

Reviewer #2:

The manuscript describes an improved method that can measure protein size at very low concentrations. This method builds on previous work developing a microfluidics system that measures molecular diffusivity. The ability to measure protein size based on the diffusivity of single proteins is a very powerful technique, and in this case, it is used to demonstrate several potential applications. While the development of this technology is exciting, the manuscript would benefit from a more detailed description of how this method compares to others, rather than just being an improved version of the previously described methods.

Reply: We thank the reviewer for the enthusiastic and insightful feedback on our manuscript. The recognition of the significant potential of our technique is greatly appreciated. Following the constructive suggestion of the reviewer, we have enhanced our manuscript with additional content in the introduction about other existing single-molecule techniques (please refer to pages 3–4). Furthermore, we have included a detailed comparison of smMDS with these methods (please refer to page 28). This expanded discussion is intended to provide a clearer and more detailed understanding of the unique capabilities and advantages of our approach.

The introduction contains little reference to similar work in the literature. Young et al (2018) used a light scattering approach that also extracted biophysical parameters from single molecules, and there are several other instances of single-molecule studies of single proteins.

Reply: Thank you for pointing out the need for additional literature references in our introduction. We understand the value of aligning our work with existing research, notably the innovative mass photometry approach by Young et al. 2018, and other pivotal single-molecule studies focused on protein analysis. In response, we have carefully incorporated these pertinent references into the introduction of our revised manuscript (please refer to page 4). This addition not only broadens the scholarly background of our study but also duly recognizes the foundational contributions of these earlier works.

While the current method has the advantage of requiring little protein, the protein does require labelling. What are the typical labelling efficiencies that are observed for different proteins? Is there any evidence that the presence of the fluorescent tag affects protein interactions? In the case of distinguishing between α -synuclein fibrils and monomers, how is the ability to detect the monomeric species in a background of fibrils enhanced but the high degree of labelling in the monomers compared with the fibrils (only 10% labelled)?

Reply: Thank you for pointing out these important aspects. Indeed, our method requires labeling with fluorescent dyes and this procedure needs to be well controlled. As elaborated in our responses to points 1, 2 and 3 of reviewer #1, we have monitored the labeling process and quantified the labeling efficiency, expressed as the degree of labeling (DOL), along with the specific protein concentrations used for labeling, the molar excess of dye used, and the protein concentration after labeling and chromatographic purification. These parameters are now summarized in a table in the Supplementary Information (Supplementary Table 3, page 9). We have also acknowledged the significance of the limitation that our method requires labeling and thus protein concentrations in the nano- or micromolar range are necessary (page 27). This can indeed be a challenge, especially when working with proteins that are scarce or not readily available in high concentrations. Also, the reviewer is right that fluorescence labels could potentially affect protein interactions. We have highlighted this limitation in the Discussion section of the revised manuscript (page 27). With regards to distinguishing between α -synuclein fibrils and monomers, we have ensured that both the fibril and monomer samples contained 10% of labeled protein. This guarantees concentration parity

between the two, enabling direct comparative analysis. For further clarification on this aspect, please refer to page 21 in our manuscript.

Fig 2c shows that 20 pM HSA appears to give an R_H of 3.29nm when measured by step scan, but 3.63 when measured by continuous scan, however, these data points appear to be reversed in Fig 2d.

Reply: Thank you for bringing this inconsistency to our attention. We re-examined our data and realized that the 20 pM HSA step scan measurement displayed in Figure 2c was initially analyzed without employing the single-molecule analysis, specifically the burst-search algorithm. Upon reanalyzing the data with this algorithm, we obtained an updated R_H value of 3.69 nm, with an error interval of [3.51, 3.88] nm. This finding has led us to revise the information presented in Figure 2c accordingly (refer to page 9). Regarding Figure 2d, we have thoroughly re-examined the values obtained from both continuous and step scans. We can confirm that the data shown in Figure 2d is accurate. The data points presented are the result of at least three independent experiments, with the figure illustrating the calculated mean and standard deviation of these repeated measurements. To further enhance clarity and ensure transparency, we have included a table with the data from Figure 2d in the Supplementary Information (Supplementary Table 1, page 9). Additionally, we have revised the caption of Figure 2d for a clearer understanding (refer to page 9).

Fig 3 shows the range of experimentally derived values for different proteins at 10 pM. In Fig 3a, the experimental R_H shown for HSA is $\sim 4.0 \pm 1$ nm. However, the equivalent data shown in Fig 2d gives an experimental value of $\sim 3.75 \pm 0.25$ nm.

Reply: Thank you for pointing out the discrepancy between Figures 3a and 2d. Upon reviewing our data, we found that the R_H value of HSA in Figure 3a was mistakenly taken from a 100 fM data set. To ensure consistency across our figures, we have now updated Figure 3a (refer to page 11) to reflect the 10 pM HSA data set. To further enhance clarity and ensure transparency, we have included a table with the data from Figure 3a in the Supplementary Material (Supplementary Table 2, page 9) that outlines the underlying data.

Given that the abstract describes "ultrasensitive sizing of proteins down to the femtomolar concentration" what is the accuracy and reproducibility of the technique for measuring protein size? The values of $\sim 4.0 \pm 1$ nm for HSA (Fig 3a) equate to a MW of 102 kDa and a range of 42-202 kDa when using the online converter at <https://www.fluidic.com/toolkit/hydrodynamic-radius-converter/> which is higher than the expected MW of 66kDa. Similarly, the value of $\sim 2.5 \pm 1$ nm for RNase (expected MW = 14 kDa) equates to a MW of 24 kDa, with a range of 5-67 kDa.

Reply: Thank you for your comment. The term "ultrasensitive" in our context refers to the capability of measuring sizes at femtomolar concentrations. This aspect was thoroughly examined in Figure 2, where we successfully determined sizes with an error margin within 10% at concentrations as low as femtomolar. We have now included these details in the manuscript for clarity. Moreover, we assessed the impact of various burst selection parameters in the single-molecule regime on size determination. Our findings, detailed in Supplementary Figure 1 (page 5), show that a broad range of parameters consistently yielded accurate size information, underscoring the method's robustness. Furthermore, the sizes we obtained for various proteins, ranging from 1–10 nm using smMDS, are consistent with literature values, both in trend and error margins as shown in Figure 3a. This correlation underscores the reliability of smMDS in single-molecule diffusivity measurements for accurate size determination of protein analytes.

HSA is also used as a model protein to show that smMDS can be used to resolve protein oligomeric states (Fig 5). In this case, the monomeric protein has an R_H of 3.46nm at 10pM, compared to a value of 3.77nm at 10pM previously described in Fig 2c. Is this difference due to a global analysis of the data in Fig 2c, compared to the deconvolution analysis in Fig 5?

Reply: Yes, the reviewer's observation is accurate. The R_H value of 3.77 nm at 10 pM presented in Figure 2c was derived from an analysis that considered all species in the sample collectively, both monomers and oligomers. In contrast, the analysis in Figure 5 allows us to discern not just the averaged size of all species present, but also to separately identify the different oligomeric states, including the monomeric form. To clarify this distinction, we have included an explanatory note in the revised manuscript (please see page 10 for this addition).

How does the proportion of monomer/dimer/trimer/tetramer compare to that determined by other methods? It is stated that "These values are in very good agreement with the expected equilibrium distribution of serum albumins(ref 70)", but the data presented in ref 70 for BSA does not have a detailed distribution of species.

Reply: Thank you for your insightful comment. In our study, we determined the relative abundances of human serum albumin species as follows: 67.9% monomer, 20.7% dimer, 7.2% trimer, and 4.2% tetramer, as illustrated in Figure 5c. While these percentages differ slightly, they broadly align with the distributions observed for other serum albumins in previous studies. Specifically, Young et al. 2018 (formerly referenced as ref. 70) reported relative abundances for bovine serum albumin of 88.63% monomer, 9.94% dimer, 1.18% trimer, and 0.25% tetramer. We acknowledge the variation and have updated our manuscript to include these comparative data for a more comprehensive analysis (please see page 17).

Are the oligomeric species of HSA in equilibrium? The methods state that the labelled protein was passed down a SEC column to separate the unbound dye, which also should have separated the oligomeric species. If the distribution is under equilibrium, it may be expected to observe a higher proportion of smaller species at lower concentrations and an increased population of trimers and tetramers at higher protein concentrations. For this (and other measurements) it may be useful to compare how smMDS compares to other methods. Techniques such as analytical ultracentrifugation can be used at pM concentrations using fluorescently labelled protein.

Reply: Thank you for your question. The oligomeric species of HSA are in equilibrium. Although the SEC column is used to remove unbound dye and might separate some oligomeric species, it does not disturb the overall equilibrium. If, for example, the SEC column removes larger oligomeric species, the equilibrium will adjust itself to form these species again. Hence, the relative proportions of each oligomeric state are expected to remain consistent. Nonetheless, it is important to note that SEC can impact the overall concentration of the species present. Consequently, we assess the concentration in terms of monomer units post-SEC to account for these variations.

In terms of investigating the interaction between HLA and W6/32, the authors compare the result of using step mode scanning compared to their previous work using continuous scanning (Schneider et al 2023). Comparison of the data suggests that the use of step mode has allowed the inclusion of an additional data point at 10 pM, but the data for the two different methods appear to be similar – perhaps more discussion could be included as to the advantages afforded by the step mode method.

Reply: Thank you for pointing out the significance of employing step scan smMDS in the study of binding interactions. The incorporation of an extra data point at 10 pM, facilitated by step scanning, may seem modest in the context of the HLA and W6/32 interaction, but it is crucial

for examining interactions with sub-nanomolar affinities. In these cases, the binding curves shift significantly to the left, necessitating a method capable of resolving these curves at lower concentration ranges. Traditional bulk assays, lacking the sensitivity for such low concentrations, thus fall short in this aspect. Step scan mode, by contrast, could provide the required resolution, highlighting its potential in delineating high-affinity interactions. This advantage, alongside the overall benefit of single-molecule step mode scanning, has been elaborated in our revised manuscript (please see page 14).

It would be useful to include more discussion regarding how the current results compare to other methods. For example, the R_H range observed for α -synuclein oligomer subpopulations is compared to a different study that used AFM (ref 70). What are the comparisons of the different analyses? AFM also detects single particles.

Reply: Thank you for your insightful feedback. In response to your suggestion, we have broadened our discussion to more effectively compare our findings with those obtained using a variety of other methods. Our comprehensive literature review, summarized in the table below, uncovered numerous studies examining α -synuclein oligomer size distributions under preparation conditions akin to ours. These studies utilized diverse techniques including atomic force microscopy (AFM), transmission electron microscopy (TEM), small-angle X-ray scattering (SAXS), and dynamic light scattering (DLS). Overall, we found that the sizes ranges reported in these studies align well with our R_H -value distribution obtained via smMDS. We have expanded our discussion in the revised manuscript to include the mentioned studies and their respective methodologies (please refer to page 18).

Publication	R_H	Technique
Chen et al.	3–16 nm	AFM
Norris et al.	12.5 nm	AFM
Lowe et al.	11–15 nm	AFM
Apetri et al.	0.7–3.8 nm	AFM
Kaufmann et al.	7 ± 1.5 nm	TEM
Pieri et al.	10 nm	TEM
Hayden et al.	5–15 nm	TEM
Wright et al.	7.5–10 nm	TEM
Paslawski et al.	7.0 ± 0.8 nm	TEM
Lorenzen et al.	4.23 ± 0.11 nm	SAXS
Giehm et al.	9 ± 2 nm	SAXS
Paslawski et al.	4–10.3 nm	SAXS
Stefanovic et al.	8.8 ± 0.3 nm	SAXS
Qin et al.	10.35 ± 0.13 nm	SAXS
Planchard et al.	2.5–5.5 nm	DLS
Fecchio et al.	11 ± 4 nm	DLS

Reviewer #3:

Reply: Thank you for your contributions and the insightful feedback on our manuscript. We greatly appreciate Nature Communications' initiative to involve Early Career Researchers in the peer review process, recognizing its importance in fostering professional growth and enhancing scientific discourse. Your involvement has been invaluable in refining our work.

Reviewer #4:

In this manuscript, Krainer et al. developed a method single-molecule microfluidic diffusional sizing (smMDS), a variant of MDS that exploits “single-molecule” fluorescence detections in a tight excitation volume to characterize purified protein/protein complexes/simple protein mixtures at pico-to-femtomolar concentrations. The authors demonstrated the application of smMDS in quantifying the hydrodynamic radii (R_h), pairwise interaction affinities (K_d), and assembly and oligomeric states of specific proteins. smMDS features innovative experimental designs and enables protein sizing characterizations at protein concentrations 5 orders of magnitude lower than what the conventional MDS is capable of characterizing. However, there are other established single-molecule techniques capable of quantifying the dimensions, interaction affinities, and oligomeric states of proteins in vitro. A prominent example (for interaction affinity and oligomeric state measurements) is single molecule mass photometry, which is based on precise measurements of light scattering of protein molecules. Another example (for dimension measurements) is super-resolution fluorescence microscopy. A comparison between these existing methods and smMDS in measuring the same protein/protein complex properties is missing in the current manuscript. Without this comparison, it is difficult for readers to evaluate the significance and unique strength of smMDS and when to apply smMDS instead of other techniques for protein characterizations. This and other comments are specified below.

Reply: Thank you for your positive feedback and the appreciation of the developments and innovations presented in our manuscript. We appreciate your suggestion to compare our smMDS method with other established single-molecule techniques such as mass photometry and super-resolution fluorescence microscopy. In response to your insightful input, we have incorporated a section in the introduction, elaborating on these methods, and have extensively discussed them in our manuscript (refer to pages 3–4 and 28). This addition aims to offer readers a clearer understanding of the unique strengths and applications of smMDS and other single-molecule methods in the context of protein characterization.

1. As summarized above, single molecule mass photometry has been shown to successfully characterize the oligomeric states of proteins in vitro as well as quantify the dissociation constant (K_d) of protein complexes (PMID: 29700264; PMID: 33068635; PMID: 34765909). It is important to discuss how smMDS compares with single molecule mass photometry in measuring these protein properties.

Reply: Thank you for bringing these mass photometry references to our attention, particularly regarding the characterization of protein oligomers and the quantification of dissociation constants. We have now incorporated these references into our manuscript both in the introduction and the discussion section (refer to pages 3–4 and 28). Additionally, we have included a discussion comparing the capabilities of smMDS with single molecule mass photometry. This addition should provide a more comprehensive understanding of how smMDS stands in relation to mass photometry and other established techniques in the field.

2. It is unclear how sensitive smMDS is in distinguishing different oligomeric states of a protein. In Figure 5a, the authors fit the fluorescence intensity distribution to a four-component model (one “asymmetrical Gaussian” distribution and three Gaussian distributions) because they have a priori knowledge about the oligomeric states of the protein HSA. The overall shape of the fluorescence intensity histogram does not show four distinct peaks. It would be helpful if the authors can show with statistical power that their model is capable predicting the four oligomeric states for HSA (rather than some other number) via Bayesian/Akaike information criterion or some other model comparison method. In other words, can they demonstrate that if they did not know that HSA had four oligomeric states, they would be able to extract that

information and the correct states? It would also strengthen the authors' conclusions if they can use smMDS to reveal the oligomeric states and their fraction in a solution for a protein whose oligomerization has not been previously characterized and then verify their smMDS findings on oligomeric states using orthogonal approaches.

Reply: Thank you for your insightful comment. The reviewer is correct in observing that the distribution in Figure 5a does not distinctly exhibit four separate peaks. However, our decision to model the fluorescence intensity distribution with four components is not solely based on prior knowledge of HSA's oligomeric states. We critically evaluated the improvement in the fit (as gauged by the sum of square errors) with each additional Gaussian component. This analysis is illustrated in the figure provided below. Our findings indicate that incorporating one skew normal distribution and each of the first three Gaussian distributions leads to a well-discernible improvement in the fit. Conversely, adding a fourth Gaussian (representing a fifth species) or any further Gaussians (additional species) does not substantially enhance the fit. This suggests that a four-species model, corresponding to the equilibrium distribution of HSA monomers, dimers, trimers, and tetramers, is the most fitting. We have included the referenced figure in the Supplementary Information as Supplementary Figure 3 (please refer to page 7) and have incorporated a mention of this observation in the revised manuscript (please refer to page 16).

Supplementary Figure 3: Goodness of fit for different fitting models applied to the burst intensity histogram shown in Figure 5a. Shown is the sum of square errors (SSE) for each model, which includes one skew distribution and (n-1) Gaussian distributions, where n denotes the total number of distributions (corresponding to the total number of species analyzed). The graph demonstrates that beyond four species (monomer to tetramer), there is no visually discernible improvement in the fit.

A related question: Which “asymmetrical Gaussian” did the authors use? There are many asymmetrical Gaussian distributions (skew normal etc.) and it would be helpful to see one line in the supplemental info listing the actual CDF/PDF they fit to.

Reply: Thank you for your query regarding the specific nature of the “asymmetric Gaussian” distribution used in our study. In our case, we utilized the skew normal distribution to introduce asymmetry in our analysis. We acknowledge the need for clarity on this matter and have therefore updated our manuscript to specifically mention the use of the skew normal distribution. This clarification can be found in the revised version of our manuscript in the figure caption of Figure 5 (page 15) and in the main text on page 16.

3. In their characterization of the nanoscale clusters of GFP-tagged TDP-43, the authors concluded that the clusters have narrowly distributed sizes, with an R_h of 121.5 ± 14.5 nm. This is an interesting finding as a phase separation mixture usually have droplet sizes more widely distributed. In this pre-phase separating regime, can the authors provide alternative evidence that verifies the uniformly distributed cluster sizes? The authors claimed that under the condition of their smMDS measurements, the clusters were undetectable by conventional microscopy. This is puzzling. Although sub-diffraction limit features “cannot be resolved” by conventional microscopy as the authors correctly pointed out, this should only mean that precise quantification of the cluster dimensions is infeasible. The clusters themselves should nevertheless be visible and detectable by microscopy. A cluster with a diameter of ~ 223 nm contains 2000 – 7000 labelled protein molecules as estimated by the authors and should be easily visible by fluorescence microscopy in principle. It is suggested that the authors clarify this point.

Reply: We appreciate your perceptive comments on our findings. Firstly, regarding the cluster size of GFP-tagged TDP-43, the reported hydrodynamic radius (now updated to $R_h = 120 \pm 10$ nm following Reviewer #1’s recommendation on significant digits) reflects the average from three independent measurements, with the standard deviation as a measure of variation. We have clarified in the figure caption that this value represents the mean \pm standard deviation (please refer to page 24). It is important to note that this standard deviation does not imply a narrow distribution of sizes. Secondly, regarding the visibility of protein clusters in wide-field microscopy, it is true that theoretically, clusters with numerous labeled proteins should be detectable. However, wide-field imaging often struggles with overwhelming signals from excess protein in the dilute phase. This high background signal from out-of-focus light masks the distinct cluster signals, rendering them virtually indistinguishable in standard wide-field epifluorescence images. Confocal fluorescence detection, utilized in smMDS, overcomes this limitation by focusing only on in-focus planes, thereby achieving nanocluster detection. smMDS leverages confocal detection together with MDS principles to accurately size nanoclusters. We have expanded our manuscript to include a more detailed explanation on this aspect (please see page 24).

4. Related to the above point, although conventional microscopy cannot resolve sub-diffraction limit features of nanoscale protein clusters, many well-established super-resolution microscopy techniques, e.g., SIM, PALM, and STORM, are capable of characterizing nanoscale features in numerous systems in vitro and in live cells. How does smMDS compare with super-resolution microscopy in characterizing the sizes of pre-phase separating protein clusters?

Reply: Thank you for your comment regarding the comparison between smMDS and super-resolution microscopy techniques in characterizing nanoscale protein clusters. Super-resolution microscopy has indeed provided pivotal insights, as seen in studies like Cisse et al. 2013 (PMID: 23828889), Pancholi et al. 2021 (PMID: 34569155), and Longfield et al. 2023 (PMID: 37949856) (now referenced on page 26). These techniques excel in visualizing and characterizing nanoclusters within cellular environments. On the other hand, smMDS uniquely combines single-molecule detection with microfluidics, offering a distinct advantage in the in-vitro analysis of nanoclusters directly in solution. The primary goal of our study was to leverage this integration to understand the biophysical attributes of protein clusters. We have added a sentence to highlight these points. This should help readers appreciate the distinct applications and strengths of smMDS in the context of biophysical protein analysis (please refer to page 26 in the manuscript).

5. *There is not a clear distinction in the background made between the advantages and shortcomings of in vitro vs live-cell measurements. As such, it leads to some relatively one-sided statements about smMDS, which could be somewhat misleading to readers. For example, the authors stated that smMDS is superior to other techniques, e.g., FCS, because the latter relies on calibration for size determination and suffers from overparameterization. Although FCS requires calibration as the authors have correctly pointed out, this current work did not show that smMDS is not prone to overparameterization (convincing statistics showing this are notably absent, as detailed in Point 2). Besides, the authors did not address the critical point that FCS and variants can be used to make size/population measurements in live cells, which is an advantage over smMDS. While buffers like PBS somewhat approximate the cellular conditions, they are far from physiological cellular environment with the presence of numerous other small molecules and macromolecules that could possibly affect the conformation, interaction behaviors, and oligomerization of a protein of interest. It is suggested that the authors directly address not only the advantages of smMDS but also its caveats to improve the quality of the paper.*

Reply: Thank you for your constructive feedback. We recognize the importance of providing a balanced perspective on smMDS. In response to your comments, we have revised both the introduction and discussion sections to present a more comprehensive comparison, acknowledging the strengths and limitations of smMDS in relation to other methods (see pages 3–4 and 27–28). We have also refined our critique of FCS, particularly regarding the overparameterization aspect, and incorporated references for further context (refer to page 4). Additionally, we have emphasized the advantage of FCS in facilitating size/population measurements in live cells, an area where smMDS has limitations (please refer to page 28).

Reviewers' Comments:

Reviewer #1:

Remarks to the Author:

The authors have been very responsive to all reviewer comments and have improved their manuscript significantly. I consider the revised manuscript suitable for publication in Nature Communications, as is.

Reviewer #2:

Remarks to the Author:

The authors have taken the opportunity to include a thorough discussion of this novel method in the context of other methods, and have clarified the presentation of the results. It is an exciting new development that will greatly aid in understanding biomolecular interactions.

Reviewer #3:

Remarks to the Author:

Reviewer #4:

Remarks to the Author:

The authors have addressed most of Reviewer #4's comments in this revision and improved the manuscript significantly. Here are a few remaining concerns.

The reviewer appreciates the authors' reply to the original comment #2. However, there is still no statistical reason why the authors should stop at four components. Why not five? Or six? The point at which they see an additional population not "substantially enhance a fit" is determined by eyeballing rather than statistics.

Another remaining issue is related to the original comment #3. In the authors' response, $R_h = 120 \pm 10$ nm reflects the average from three independent measurements, with the standard deviation as a measure of variation. This is different from what the manuscript states, "we obtained a R_h of 120 ± 10 nm for TDP-43 nanoclusters". It will be misleading to readers if the standard deviation of the hydrodynamic radius (10nm) is not calculated from the individual measurements of 43 nanoclusters. This inconsistency needs to be addressed. The standard deviation needs to be calculated from the individual measurements of 43 nanoclusters.

Response to reviewers' comments

Reviewer #1:

The authors have been very responsive to all reviewer comments and have improved their manuscript significantly. I consider the revised manuscript suitable for publication in Nature Communications, as is.

Reply: Thank you for the positive feedback and for considering our manuscript suitable for publication as is. We appreciate the reviewer's thoughtful comments and guidance throughout the revision process.

Reviewer #2:

The authors have taken the opportunity to include a thorough discussion of this novel method in the context of other methods, and have clarified the presentation of the results. It is an exciting new development that will greatly aid in understanding biomolecular interactions.

Reply: Thank you for recognizing the enhancements to our manuscript, particularly the detailed discussion of smMDS in relation to existing techniques and the clearer presentation of our results. We appreciate your enthusiasm for the potential impact of our work on the understanding of biomolecular interactions.

Reviewer #3:

Reply: Thank you for your support in reviewing our manuscript. We appreciate your thoughtful suggestions, which have been instrumental in improving our manuscript.

Reviewer #4:

The authors have addressed most of Reviewer #4's comments in this revision and improved the manuscript significantly. Here are a few remaining concerns.

Reply: Thank you for your thorough review and valuable feedback on our manuscript. Below we have prepared a point-by-point response to address the reviewer's remaining comments and valuable suggestions.

The reviewer appreciates the authors' reply to the original comment #2. However, there is still no statistical reason why the authors should stop at four components. Why not five? Or six? The point at which they see an additional population not "substantially enhance a fit" is determined by eyeballing rather than statistics.

Reply: Thank you for revisiting this point. Initially, our assessment was indeed based on a qualitative evaluation of different fitting models. In response to the reviewer's

suggestion, we have now applied the Bayesian Information Criterion (BIC) and the Akaike Information Criterion (AIC) to our data (see figure below). In this analysis, we calculated ΔBIC and ΔAIC scores as the differences between the BIC and AIC scores for each model and the model with the lowest BIC or AIC scores, respectively. Our analysis indicates that incorporating one skew normal distribution and three Gaussian distributions in the model presents the lowest ΔBIC and ΔAIC scores, substantiating that a four species model, corresponding to the equilibrium distribution of HSA monomers, dimers, trimers, and tetramers, is the most fitting. We note that for the BIC analysis, a five species model already gives a ΔBIC score of >5 and any higher-species model gives ΔBIC scores of >10 . According to the criteria set forth by Raftery (1995),¹ a BIC difference of 2 to 6 suggests ‘positive’ evidence in favor of the model with the smaller BIC, and a difference of 6 to 10 suggests ‘strong’ evidence for that model. For the AIC analysis, we obtain ΔAIC scores of <2 for the five and six species models, and higher order models give scores >2 . Here, according to Burnham and Anderson (2004),² ΔAIC scores <2 are considered to be essentially as good as the best-fitting model, whereas models with greater scores have considerably less or no support. Accordingly, the analysis based on ΔAIC scores also suggest that a four-species model is the most fitting, however, it cannot entirely rule out a five or six species model. We have included the referenced figure in the Supplementary Information as Supplementary Figure 3 (along with Supplementary Note 2, please refer to pages 4 and 7) and have incorporated a mention of this observation in the revised manuscript (please refer to page 12).

Supplementary Figure 3: Statistical analysis of various fitting models applied to the burst intensity histogram shown in Figure 5a. Displayed are the differences (ΔBIC or ΔAIC) in Bayesian Information Criterion (BIC) and Akaike Information Criterion (AIC) scores for each model relative to the model with the lowest BIC or AIC value, respectively. Each model includes one skew normal distribution and $(n-1)$ Gaussian distributions, where n represents the total number of distributions (corresponding to the total number of species analyzed). The lowest ΔBIC and ΔAIC scores were found for the four-species model. The AIC and BIC scores were calculated from the sum of square errors (SSE) as follows, with k denoting the complexity of the model and n denoting the number of points to fit: $\text{AIC} = 2k - 2 \ln(\text{SSE}/n)$ and $\text{BIC} = k \ln(n) - 2 \ln(\text{SSE}/n)$.

Another remaining issue is related to the original comment #3. In the authors' response, $R_h = 120 \pm 10$ nm reflects the average from three independent measurements, with the standard deviation as a measure of variation. This is different from what the manuscript states, "we obtained a R_h of 120 ± 10 nm for TDP-43 nanoclusters". It will be misleading to readers if the standard deviation of the hydrodynamic radius (10nm) is not calculated from the individual measurements of 43 nanoclusters. This inconsistency needs to be addressed. The standard deviation needs to be calculated from the individual measurements of 43 nanoclusters.

Reply: Thank you for your comment. We apologize for any confusion caused. To clarify, we have revised the relevant sentence in our manuscript to now state: "We performed this analysis with three independent measurements and obtained an average R_H of 120 ± 10 nm for TDP-43 nanoclusters (Figure 8f)." Additionally, we have updated the caption of Figure 8f to read: "The extracted average R_H of TDP-43 nanoclusters is given as an inset (mean \pm standard deviation)."

References:

1. Raftery, A. E. Bayesian Model Selection in Social Research. *Sociol Methodol* **25**, 111 (1995).
2. Burnham, K. P. & Anderson, D. R. *Model Selection and Multimodel Inference: A Practical Information-Theoretic Approach*. (Springer New York, New York, NY, 2002). doi:10.1007/b97636.